EMBO
Molecular Medicine

# Unraveling autophagic imbalances and therapeutic insights in Mecp2-deficient models

Alessandro Esposito [1✉], Tommaso Seri [1], Martina Breccia[2], Marzia Indrigo[1], Giuseppina De Rocco [1,2], Francesca Nuzzolillo[3], Vanna Denti [4], Francesca Pappacena [1], Gaia Tartaglione[1], Simone Serrao [4], Giuseppe Paglia[4], Luca Murru [5], Stefano de Pretis[6], Jean-Michel Cioni[1], Nicoletta Landsberger [1,2], Fabrizia Claudia Guarnieri [1,5✉] & Michela Palmieri [1,3✉]

## Abstract

Loss-of-function mutations in *MECP2* are associated to Rett syndrome (RTT), a severe neurodevelopmental disease. Mainly working as a transcriptional regulator, MeCP2 absence leads to gene expression perturbations resulting in deficits of synaptic function and neuronal activity. In addition, RTT patients and mouse models suffer from a complex metabolic syndrome, suggesting that related cellular pathways might contribute to neuropathogenesis. Along this line, autophagy is critical in sustaining developing neuron homeostasis by breaking down dysfunctional proteins, lipids, and organelles.

Here, we investigated the autophagic pathway in RTT and found reduced content of autophagic vacuoles in *Mecp2* knock-out neurons. This correlates with defective lipidation of LC3B, probably caused by a deficiency of the autophagic membrane lipid phosphatidylethanolamine. The administration of the autophagy inducer trehalose recovers LC3B lipidation, autophagosomes content in knock-out neurons, and ameliorates their morphology, neuronal activity and synaptic ultrastructure. Moreover, we provide evidence for attenuation of motor and exploratory impairment in *Mecp2* knock-out mice upon trehalose administration. Overall, our findings open new perspectives for neurodevelopmental disorders therapies based on the concept of autophagy modulation.

**Keywords** Autophagy; MeCP2; Metabolism; Neurons; Rett Syndrome
**Subject Categories** Autophagy & Cell Death; Neuroscience

## Introduction

While most of the cells in our body can divide and easily remove unwanted materials, post-mitotic and long-living cells, such as neurons, require integrated degradative processes to prevent the accumulation of cellular debris and sustain their survival (Stavoe and Holzbaur, 2019). Macroautophagy (hereafter referred to as autophagy) is a highly conserved catabolic process deputed to the elimination of dysfunctional proteins, lipids, and organelles (Dikic and Elazar, 2018). It is fundamental in neuronal cells for managing nutrient content, energy resources, and cellular waste. Autophagy is tightly regulated and consists of four sequential steps: initiation, phagophore formation, membrane expansion, and autophagosome maturation. The initiation step is rather complex and requires dozens of factors; among them, the mammalian target of rapamycin complex 1 (mTORC1) is undoubtedly a key modulator of autophagy function that works in response to nutrient supply. When nutrients are available, mTORC1 gets activated and phosphorylates components of the autophagy initiation machinery or ULK-complex (ULK1/2, ATG13, FIP200, and ATG101), thus inhibiting the process. When nutrients are scarce, mTORC1 is inactivated and allows the recruitment of the nucleation complex (Beclin1, Vps34, Vps15, ATG14, and WIPI) on the forming phagophore. Subsequently, the microtubule-associated protein 1 light chain 3 (LC3) lipidation machinery mediates the proteolytic cleavage of LC3 and its conjugation to phosphatidylethanolamine (PE) on the nascent double-membrane of the phagophore. This cascade of events is coordinated by several autophagy-related genes (ATGs). Firstly, ATG4 cleaves LC3 to originate LC3-I, which is then activated by ATG7 (E1-like enzyme) and transferred to ATG3 (E2-like enzyme) and ATG12-ATG5-ATG16L1 complex (E3 ligase) for PE conjugation (Bingol, 2018). The newly formed LC3-II participates in cargo selection and promotes phagophore expansion and sealing. The mature autophagosome finally tethers and fuses to the lysosome, thus forming the autolysosome, a key organelle deputed to the degradation of engulfed cellular components (Hill and Colón-Ramos, 2020).

Dysfunctional neuronal autophagy has been extensively described during aging and in neurodegenerative diseases (Fleming et al, 2022). More recently, mutations in genes encoding for components of the autophagic-lysosomal system have been identified in neurodevelopmental disorders and childhood-onset neurological diseases (Ebrahimi-Fakhari et al, 2016; Fassio et al, 2020; Guerrini et al, 2022; Zhu et al,

[1]Division of Neuroscience, IRCCS San Raffaele Scientific Institute, Milan, Italy. [2]Department of Medical Biotechnology and Translational Medicine, University of Milan, Segrate, Italy. [3]Vita-Salute San Raffaele University, Milan, Italy. [4]School of Medicine and Surgery, University of Milano-Bicocca, Milan, Italy. [5]CNR Institute of Neuroscience, Vedano al Lambro, Italy. [6]Center for Omics Sciences, IRCCS San Raffaele Scientific Institute, Milan, Italy. ✉E-mail: esposito.alessandro@hsr.it; fabrizia.guarnieri@in.cnr.it; palmieri.michela@hsr.it

2019). Accordingly, loss or mutation of essential autophagic components and regulators have been associated in humans with developmental delay and seizures, and with the impairment of neurite extension in cultured neuronal models (Esposito et al, 2019; Fassio et al, 2018; Kadir et al, 2016). In addition, the removal of core autophagy genes in rodent neurons leads to defects in synapse maturation, synaptic assembly, neurotransmitter release and neuronal plasticity (Hernandez et al, 2012; Kuijpers and Haucke, 2021; Nikoletopoulou et al, 2017; Overhoff et al, 2022; Tang et al, 2014). Indeed, autophagy regulates the levels of several synaptic components, including synaptic vesicles, postsynaptic receptors, and scaffolds (Compans et al, 2021; Goldsmith et al, 2022; Kallergi et al, 2022, 2023; Fleming and Rubinsztein, 2020).

Loss of function mutations in the X-linked methyl-binding protein 2 (*MECP2*) gene are causative of Rett syndrome (RTT), a devastating neurodevelopmental disorder that mostly affects females. After an apparent normal development that lasts 6 to 18 months, RTT girls experience a regression phase characterized by the onset of a variety of neurological features, including motor impairments, loss of acquired language, intellectual disability, anxiety and seizures (Amir et al, 1999; Chahrour and Zoghbi, 2007). *MECP2* encodes for an epigenetic factor mainly working as a global transcriptional regulator (Chahrour et al, 2008). Accordingly, its loss results in broad perturbation of gene expression that in neurons leads to developmental delay, with a reduction in cell size and dendritic arborization, altered circuit connectivity, and defective synaptic plasticity (Feldman et al, 2016). These features are well recapitulated by *Mecp2* knock-out (KO) mouse models (Guy et al, 2001). Besides these very well-characterized neurological phenotypes, RTT patients and mouse models are now extensively recognized to suffer from a complex metabolic syndrome, with variable presentation (Kyle et al, 2018). This includes variations in glucose metabolism, insulin resistance, severe dyslipidemia, and altered brain cholesterol levels that are likely to impact the metabolic status of neuronal cells, and thus on autophagic activity. Accordingly, defects in mTOR signaling have been described in RTT brains and *Mecp2* KO mice (Olson et al, 2018; Ricciardi et al, 2011), and defective autophagy has been reported in RTT skin fibroblasts (Sbardella et al, 2017). Interestingly, electron microscopy data back in the '90 s indicated the accumulation of undegraded materials in the brain of RTT patients (Cornford et al, 1994; Papadimitriou et al, 1988); however, no further studies addressed the origin of altered autophagy especially in brain tissues.

Here, we investigated the autophagic pathway in *Mecp2* KO neurons and found a reduced content of autophagic vacuoles and defective LC3B lipidation likely due to PE deficiency leading to impaired autophagosome maturation. By inducing the autophagic pathway with the disaccharide trehalose, we enhanced the biogenesis of these organelles and ameliorated the neuronal features and locomotor-exploratory behavior of *Mecp2* KO mice.

# Results

## The biogenesis of autophagosomes is defective in *Mecp2* KO neurons

Based on literature data indicating the accumulation of undegraded material in the brain of RTT patients (Cornford et al, 1994;

Papadimitriou et al, 1988), we wanted to investigate whether the autophagic pathway is perturbed in neuronal models lacking Mecp2. To do so, we analyzed the protein levels of LC3B-I, LC3B-II and p62, typically used as readouts of autophagic function (Bjørkøy et al, 2009), in cultured cortical neurons generated from *Mecp2* KO and wild-type (WT) embryos. As shown in Figs. 1A and EV1A, KO neurons displayed a significant decrease in the LC3B-II/LC3B-I ratio when compared to WT cells at 14 days in vitro (DIV), while no significant alteration was observed at earlier developmental stages (Fig. EV1A). Previous work demonstrated that *Mecp2* KO neurons suffer from a defective degree of maturation compared to WT neurons, but they do not fail to progress into differentiation steps (Scaramuzza et al, 2021; Frasca et al, 2020). To confirm that the LC3B-II lipidation defect was not due to a shift in the differentiation stage of KO neurons, we analyzed the synaptic markers VAMP2, PSD-95, and SNAP25, along with NeuN as a marker of post-mitotic differentiation. These markers display a similar pattern of expression along time in WT and KO neurons (i.e., they peak at the same time point), thus suggesting that we are comparing similar developmental windows in the two genotypes (Fig. EV1B).

To validate these data in vivo, we isolated cortices from WT and *Mecp2* KO mice at postnatal day 5 (P5) and found the same defect in the LC3B-II/LC3B-I ratio (Fig. 1B). In both experimental models, the autophagic receptor p62 was not significantly different in the two genotypes (Fig. 1A,B). In addition, in line with previous findings (Sbardella et al, 2017), we also found a lower amount of LC3B-II and accumulation of p62 in skin fibroblasts of an RTT patient carrying the *MECP2* c.705delG mutation (p.E235fs) compared to healthy control cells (Fig. 1C).

To ascertain whether this defect corresponds to a lower number of autophagic vesicles, we performed transmission electron microscopy (TEM) analysis. Electron micrographs evidenced that autophagic vacuoles (autophagosomes and autolysosomes) occupied a reduced surface of the soma in KO cultured neurons with respect to WT, with no significant difference in lysosomal surface area (Fig. 1D). Furthermore, in a few cases, we observed an anomalous presence of immature autophagic structures in KO neurons that were never found in WT cells. These structures appeared as early-stage double membranes starting to engulf cytoplasmic material and mitochondria, failing to end up in the generation of a mature autophagosome (Fig. EV1C).

Next, we assessed the autophagic flux in WT and KO neurons transfected with the tandem mCherry-EGFP-LC3 construct, a well-established autophagic flux reporter (Klionsky et al, 2016). We observed a significant decrease in the total number of yellow puncta in the cell bodies of KO neurons, while the percentage of yellow over total red puncta was unaffected with respect to WT neurons (Fig. 1E). Accordingly, the transient overexpression of the same plasmid in fibroblasts of an RTT patient, described above, confirmed the reduced number of yellow puncta and the unaltered yellow over red dots ratio (Fig. 1F). These data indicate a defective autophagosome content with preserved lysosomal fusion in cells lacking MeCP2.

Compelling evidence demonstrated that neuronal autophagy, under basal conditions, is fundamental for the turnover of numerous synaptic substrates (Goldsmith et al, 2022; Kallergi et al, 2023). For this reason, we measured the levels of the vesicular glutamate transporter 1 (VGLUT1) as a readout of basal autophagy of synaptic targets and found increased levels of this protein in *Mecp2* KO neurons as compared to WT (Fig. EV1D).

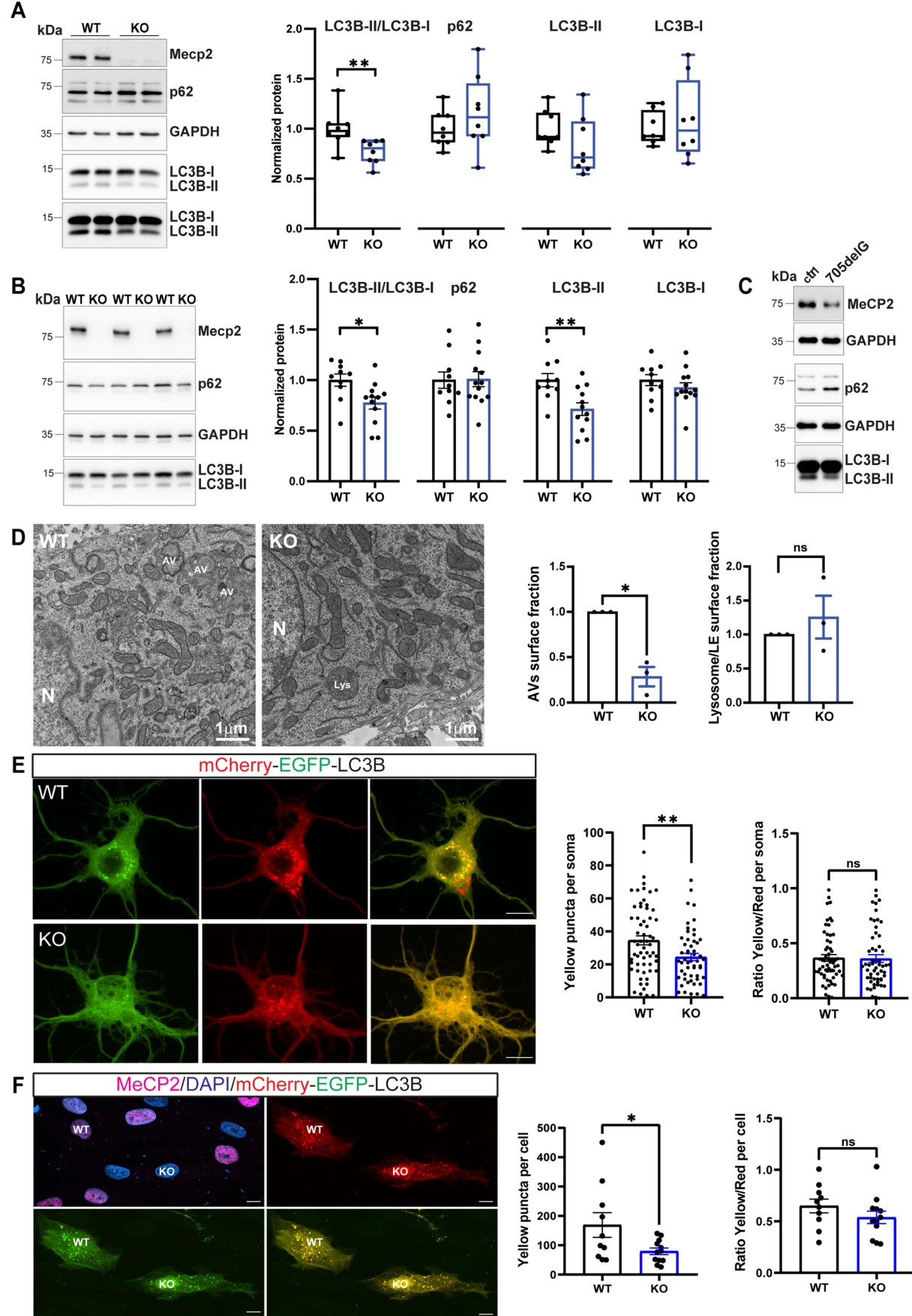

**Figure 1. Mecp2 deficiency leads to a reduction in LC3-II levels in vitro and in vivo.**

(A) Representative western blot from lysates of WT and KO cortical neurons (14 days in culture, DIV). p62, LC3B-I, and LC3B-II intensities were quantified by densitometric analysis and normalized on GAPDH intensity. LC3B-II/LC3B-I ratio was also calculated. Mecp2 signal is shown as a genotype control. Data were expressed as median ± min/max, normalized on WT ($n = 8$ embryos from three independent experiments). Mann–Whitney test, **$p < 0.01$ (LC3B-II/LC3B-I: $p = 0.003$; p62: $p = 0.2786$; LC3B-II: $p = 0.1605$; LC3B-I: $p = 0.9591$). (B) Protein expression analysis from cortices of WT and KO mice at P5 are shown. The band densities of LC3B-I-II, and p62 are quantified and normalized over GAPDH. Data were expressed as mean ± SEM normalized on WT ($n = 9$ WT mice; $n = 12$ KO mice). Unpaired $t$-test with Welch's correction, *$p < 0.05$, **$p < 0.01$ (LC3B-II/LC3B-I $p = 0.0165$, p62/GAPDH $p = 0.9318$, LC3B-II/GAPDH $p = 0.0048$, LC3B-I/GAPDH $p = 0.3241$). Outlier was identified with GRUBBS's test and removed from the analysis. (C) Western blot analysis was performed on primary fibroblast lysates from an RTT patient carrying a mutation in *MECP2* (c.705delG; p.E235fs) and a healthy control (ctrl). MeCP2 expression levels, as well as p62, LC3B-I and LC3B-II, are shown. GAPDH was used as a loading control ($n = 1$ experiment). (D) Representative TEM micrographs of WT and KO cortical neurons (14 DIV) under resting conditions. The area occupied by autophagosomes and autolysosomes (AVs) or by lysosomes over the total cell area in the image was quantified. Data were expressed as mean ± SEM ($n = 3$ independent experiments, with 59 WT and 60 KO neurons analyzed). Unpaired $t$-test with Welch's correction, *$p < 0.05$ (AVs $p = 0.0212$; Lysosomes $p = 0.5013$). N nucleus, AV autophagosome, Lys lysosome. (E) Representative images of WT and KO cortical neurons (14 DIV) transfected with the mCherry-EGFP-LC3 construct. The total number of yellow dots in the cell soma and the ratio of yellow/red dots were quantified. Data were expressed as mean ± SEM ($n = 4$ independent experiments, with 60 WT and 53 KO neurons analyzed). Unpaired $t$-test with Welch's correction, **$p < 0.01$ (Yellow puncta per soma: $p = 0.0053$, Ratio Yellow/Red per soma: $p = 0.8946$). (F) Representative images of RTT patients' fibroblast transfected with the mCherry-EGFP-LC3 plasmid. The total number of yellow puncta and the yellow/red dots ratio were quantified. Data were expressed as mean ± SEM ($n = 2$ independent experiments, with 10 WT and 12 KO analyzed fibroblasts). Unpaired $t$-test, *$p < 0.05$ (Yellow puncta per cell: $p = 0.0370$, Ratio Yellow/Red per cell: $p = 0.2336$). Source data are available online for this figure.

These results led us to hypothesize that the biogenesis of autophagosomes is reduced in the absence of Mecp2. To gain more insight into the status of the autophagic flux, WT and KO cultured neurons were treated with leupeptin, which inhibits lysosomal hydrolases and blocks the clearance of lysosomal content, including LC3-II and p62 derived from autophagosome-lysosome fusion (Boland et al, 2008). As expected, at the ultrastructural level, leupeptin treatment (20 μM for 24 h) induced the appearance of double-membrane organelles containing electron-dense material in the cell bodies and neurites, in both genotypes (Appendix Fig. S1A), similarly to what has been previously described (Boland et al, 2008). As shown in Fig. 2A, leupeptin treatment caused the accumulation of LC3B-II and p62, measured by western blot, in both KO and WT neurons as compared to untreated cells, indicating that basal degradation of the autophagic content correctly occurs in neurons lacking Mecp2. However, after leupeptin treatment, the LC3B-II/LC3B-I ratio was still lower in KO neurons than in WT cells, suggesting a defect in the early steps of autophagosome maturation rather than in the late steps of lysosomal degradation. In line with that, we did not find significant differences in the protein expression of the lysosomal marker LAMP1 between the two genotypes (Appendix Fig. S1B).

These results were further confirmed when LC3B-II levels were analyzed by western blot in neurons subjected to nutrient stimuli known to modulate the autophagic pathway. Amino acid-rich conditions stimulate mTORC1 activity to promote anabolic processes, while amino acid depletion inactivates mTORC1 and enhances autophagy (Demetriades et al, 2014). In WT neurons, 4 h of amino acid starvation enhanced autophagic flux, as evidenced by a reduction in LC3B-II levels. One hour of amino acid refeeding restored LC3B-II levels, as expected. In KO neurons, both amino acid starvation and refeeding did not induce any significant variation in LC3B-II levels (Fig. 2B). The analysis of mTORC1 activity through the phosphorylation status (Thr389) of its direct substrate p70S6 kinase 1 (p70S6K1) in fed, starved and refed conditions showed that both WT and KO neurons responded to these stimuli, even though with subtle differences between the two genotypes (Fig. 2C). This indicates that the unresponsiveness of LC3B-II levels upon variations in nutrient availability is not dependent on a defect in the nutrient-sensing ability of Mecp2-lacking neurons.

Together, these data suggest that the biogenesis of new autophagosomes is impaired in *Mecp2* KO neurons, and this relates to a defect in the conversion of LC3B-I to LC3B-II.

## Unbalanced lipid content impacts LC3B lipidation in *Mecp2* KO neurons

To understand the cause of the impaired conversion of LC3B-I to LC3B-II in *Mecp2* KO neurons, we evaluated the expression levels of core components of the LC3 lipidation machinery. The levels of ATG3, ATG5, and ATG16L1β were unaffected in KO neurons as compared to WT, while we detected a significant decrease in ATG16L1α that may contribute to a lower lipidation of LC3B (Fig. 3A). Both ATG16L1 α and β isoforms have been described to recruit the ATG12-ATG5 complex at phagophore and indicate the site for LC3 lipidation; however, the role of the specific isoform on this process is still unclear (Fujita et al, 2008). LC3 lipidation also critically depends on the availability of phosphatidylethanolamine (PE) on the nascent autophagosomal membrane (Nebauer et al, 2007). Importantly, lipid metabolism is dramatically altered in RTT patients and *Mecp2* KO mouse models (Buchovecky et al, 2013; Golubiani et al, 2021; Kyle et al, 2016, 2018). We thus performed lipidomics analysis through mass spectrometry in *Mecp2* KO and WT neurons at 14 DIV. Interestingly, the levels of PE species were significantly under-represented in KO neurons compared to WT (Fig. 3B). Given that the availability of PE is a limiting step in the process of autophagic elongation (Rockenfeller et al, 2015), the scarceness of this lipid may substantially impact the rate of LC3 lipidation and autophagosome maturation in *Mecp2* KO neurons. Accordingly, ethanolamine supplementation in the cell medium restored the LC3B lipidation capacity in *Mecp2* KO neurons (Appendix Fig. S2A,B).

## Trehalose administration ameliorates autophagy-related phenotypes in *Mecp2* KO neurons

Among the known autophagy inducers, the natural non-reducing disaccharide trehalose is a powerful activator both in vitro and in vivo (Castillo et al, 2013; Palmieri et al, 2017; Rodríguez-Navarro et al, 2010; Sarkar et al, 2007; Tanaka et al, 2004; Zhang et al, 2014).

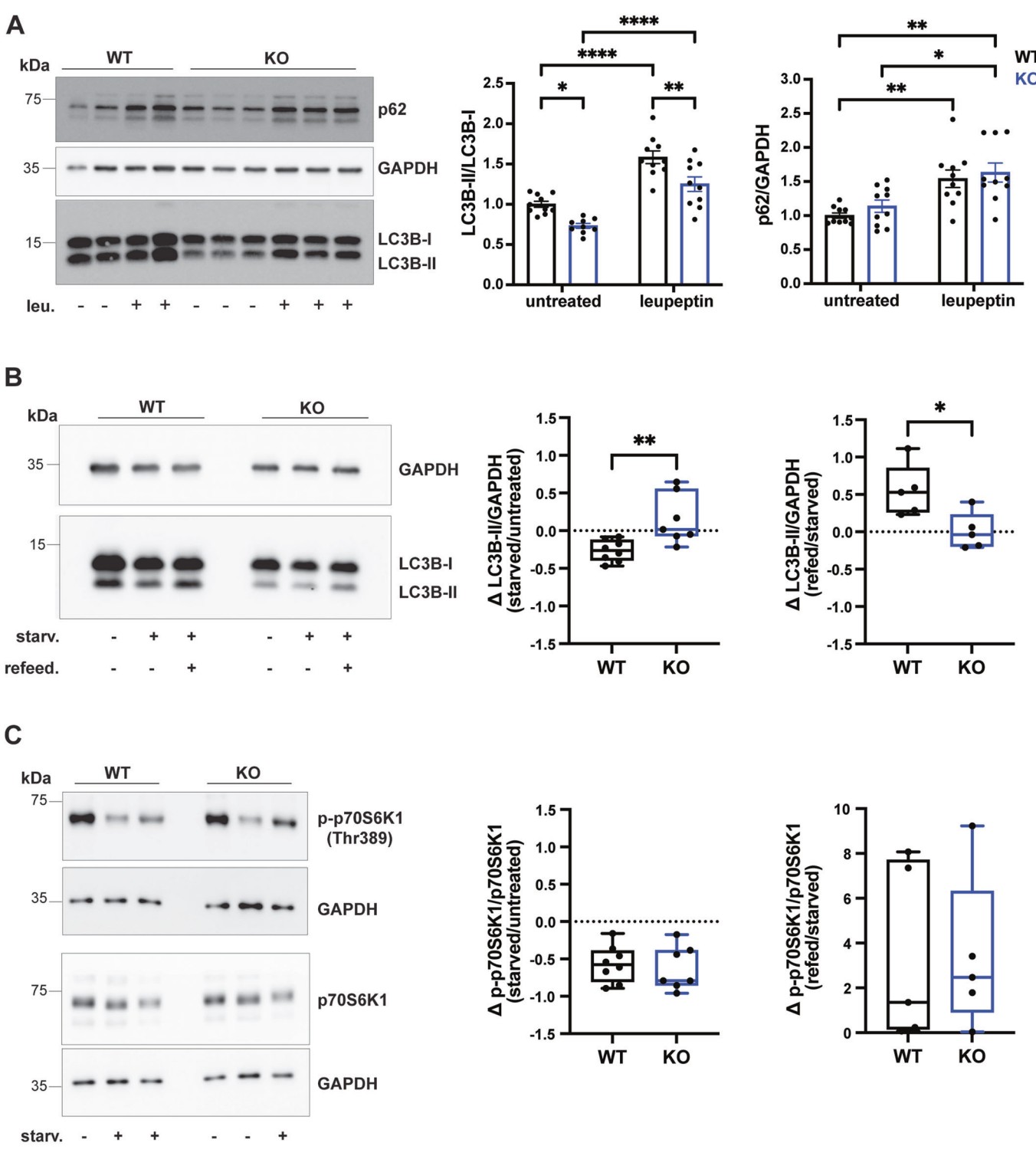

Accordingly, previous data demonstrated that trehalose induces the de novo expression of LC3B and promotes its lipidation to the LC3B-II form (Palmieri et al, 2017; Sarkar et al, 2007). To test whether this disaccharide could restore the levels of LC3B-II in neuronal cells, WT and *Mecp2* KO cortical neurons were treated or not with 25 mM trehalose at 12 DIV. After 48 h of treatment,

western blot analysis showed that LC3B-II was induced to a similar level in both genotypes, thus abolishing the basal difference in LC3B-II/LC3B-I ratio (Fig. 4A). At the ultrastructural level, trehalose treatment promoted the appearance of abundant clear-content compartments of putative lysosomal identity and of complex electron-dense lamellated autophagic organelles in cell

**Figure 2. LC3B-II low levels do not depend on a defect in the nutrient-sensing ability of *Mecp2* KO neurons.**

(A) Representative western blot from lysates of WT and KO cortical neurons (14 DIV) under resting condition or incubated with 20 μM leupeptin for 24 h. p62, LC3B-I, and LC3B-II intensities were quantified by densitometric analysis. LC3B-II/LC3B-I ratio was calculated, and the p62 level was normalized on GAPDH intensity. Data were expressed as mean ± SEM, normalized on WT untreated ($n = 9$–10 embryos from four independent experiments). Two-way ANOVA with Tukey's multiple comparisons test, *$p < 0.05$, **$p < 0.01$, ****$p < 0.0001$ (LC3B-II/I: ut WT vs ut KO $p = 0.0355$, ut WT vs leu WT $p < 0.0001$, ut WT vs leu KO $p = 0.0507$, ut KO vs leu WT $p < 0.0001$, ut KO vs leu KO $p < 0.0001$, leu WT vs leu KO $p = 0.0047$; p62/GAPDH: ut WT vs ut KO $p = 0.7996$, ut WT vs leu WT $p = 0.0055$, ut WT vs leu KO $p = 0.0010$, ut KO vs leu WT $p = 0.0545$, ut KO vs leu KO $p = 0.0126$, leu WT vs leu KO $p = 0.9313$). (B) Representative western blot from lysates of WT and KO cortical neurons (14 DIV) under resting condition (untreated), withdrawal of amino acids (starvation), and subsequent recovery by replacement of starvation medium with normal cell growth medium (refeeding). LC3B-II intensity was quantified by densitometric analysis and normalized on GAPDH intensity and on WT untreated. The fold change (delta, Δ) of LC3B-II intensity was then calculated as (starvation-untreated)/untreated or (refeeding-starvation)/starvation. Data were expressed as median ± min/max ($n = 5$–8 embryos from five independent experiments). Mann–Whitney test, *$p < 0.05$, **$p < 0.01$ (Δ LC3B-II starved/ut: $p = 0.0022$; Δ LC3B-II refed/ut: $p = 0.0317$). (C) Representative western blot from lysates of WT and KO cortical neurons (14 DIV) under resting condition (untreated), starvation and refeeding conditions as in (B). p70S6K1 and p-p70S6K1 (Thr389) intensities were quantified by densitometric analysis and normalized on GAPDH intensity and on WT untreated. The fold change (delta, Δ) of p-p70S6K1/p70S6K1 ratio was then calculated as (starvation-untreated)/untreated or (refeeding-starvation)/starvation. Data were expressed as median ± min/max ($n = 5$–8 embryos from five independent experiments). Mann–Whitney test, ns (Δ p-p70S6K1 starved/ut: $p = 0.6943$; Δ p-p70S6K1 refed/ut: $p = 0.8413$). Source data are available online for this figure.

bodies and neurites in both WT and KO neurons, thus suppressing the basal difference in the autophagic surface fraction observed between the two genotypes (Fig. 4B). Interestingly, lipidomic analyses performed by mass spectrometry revealed a substantial increase, particularly in the levels of PE species in KO neurons treated with trehalose when compared to untreated KO or WT cells (Figs. 4C,D and EV2A). The ATG16L1α levels were not recovered in KO neurons (Fig. EV2B), thus suggesting that PE deficiency might be the primary cause of defective LC3B lipidation. To further explore the molecular pathways induced by this disaccharide in WT and *Mecp2* KO neurons, we performed bulk RNA-seq on trehalose-treated and untreated samples at 14 DIV as described above. PCA analysis indicated a clear separation between WT and KO neurons, while KO-treated neurons are more proximal to WTs (Fig. EV3A). Expression change analyses revealed that a total of 390 (290 UP and 100 DW) and 447 (281 UP and 166 DW) genes were found altered by trehalose in WT and KO neurons, respectively (Fig. EV3B,C). Remarkably, DEGs with log2 fold change >0.25 in treated neurons compared to untreated controls underlined the emergence of biological pathways related to lipid metabolism as the most significantly modified by trehalose treatment in both genotypes (Fig. 5A,B and Appendix Tables S1,2). However, detailed analyses of genes exclusively upregulated in either KO ($n = 130$) or WT ($n = 139$) samples as indicated by the Venn diagram (Fig. 5C) evidenced a strong enrichment in membrane lipid and sphingolipid metabolic processes in KO neurons (Fig. 5D and Appendix Table S3). In contrast, cell proliferation and cellular metabolic pathways are overrepresented in WT-treated neurons (Fig. 5D). Remarkably, among the list of downregulated genes in untreated KO neurons compared to untreated WT cells, we found several of them related to metabolic functions (Appendix Table S4). Finally, in agreement with previous studies (Palmieri et al, 2017; Rusmini et al, 2019), the expression changes in WT and KO treated neurons versus their untreated controls confirmed the ability of trehalose to activate the lysosomal function in both genotypes (Fig. 5A,B). Detailed gene set enrichment analysis (GSEA) of autophagy-lysosomal genes further demonstrated their over-representation among the list of upregulated genes in the RNA-seq dataset (Fig. 5E,F). In line with these results, we verified the expected subcellular redistribution of TFEB in WT and *Mecp2* KO neurons expressing EGFP-TFEB upon trehalose administration. Confocal microscopy analysis revealed that TFEB translocated from the

cytosol to the nucleus to a similar extent in the two genotypes (Fig. EV3D) after 48 h of treatment.

All together these data indicate that trehalose treatment reverts the main autophagy-related alterations observed in cultured *Mecp2* KO neurons, through transcriptional effects that correlate with metabolic changes.

## Autophagy enhancement improves neuronal defects of *Mecp2* KO neurons

Prompted by the metabolic recovery of *Mecp2* KO neurons, we tested whether trehalose administration could also ameliorate neurodevelopmental features associated with *Mecp2* deficiency, including defective neurite branching, neuronal activity, and synapse organization (Frasca et al, 2020; Scaramuzza et al, 2021). WT and KO neurons were treated with trehalose for 48 h and morphologically assessed by Sholl analysis. This analysis proved that trehalose significantly increased dendritic branching and recovered the defective neuronal morphology of *Mecp2* KO neurons (Fig. 6A and Appendix Fig. S3A). We then tested whether trehalose administration also improved the responsiveness of *Mecp2* KO neurons to glutamatergic stimuli. In line with previous observations (Scaramuzza et al, 2021), the analysis of intracellular Ca²⁺ transients induced by 100 μM NMDA indicated an impaired responsiveness in neurons lacking *Mecp2* at 14 DIV. Trehalose administration partially normalized their ability to respond to NMDA (Fig. 6B). In addition, we analyzed the effects of trehalose on synaptic ultrastructure by TEM. Interestingly, we found that *Mecp2* KO neurons display a higher density of synaptic vesicles in the presynaptic terminal, as well as an increased density of vesicles docked to the active zone as compared to WT neurons, with no alteration in the presynaptic terminal area (Fig. 6C and Appendix Fig. S3B). This is consistent with the above-mentioned increase in VGLUT1 protein levels in KO cells (Fig. EV1D). Trehalose treatment restored synaptic vesicle density in KO neurons to WT levels (Fig. 6C). Unfortunately, this was not accompanied by a restoration of synaptic activity assessed by patch-clamp recording of miniature excitatory postsynaptic currents (mEPSCs). Indeed, KO neurons presented an increase in the frequency and amplitude of mEPSCs, as previously reported (Li et al, 2016), that was not modified by trehalose treatment (Appendix Fig. S3C).

## A

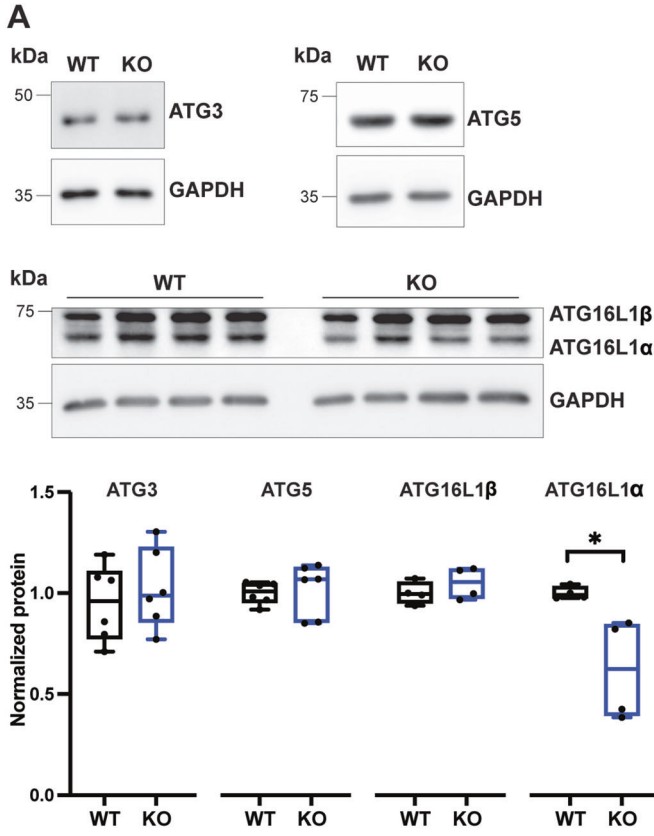

## B

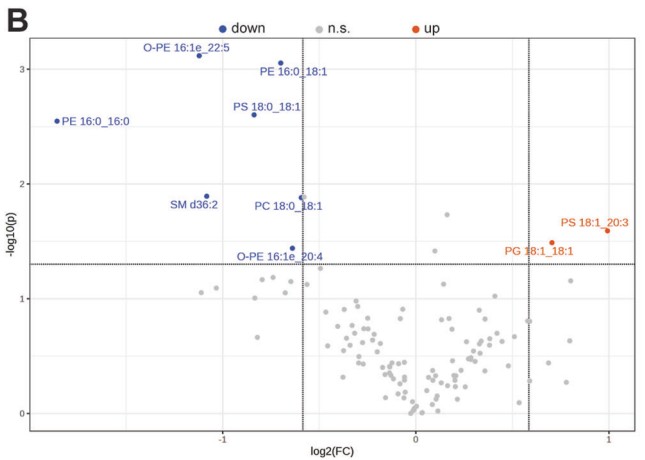

To expand our in vitro findings, we employed another autophagy inducer, the mTOR inhibitor rapamycin, and assessed the recovery of autophagosome content and neuronal defects in *Mecp2* KO neurons. Similarly to trehalose data, rapamycin treatment increased the autophagic vesicles to a larger extent in KO than WT treated neurons (Fig. EV4A); moreover, it also stimulated the neuronal activity in *Mecp2* deficient neurons upon 48 h of administration (Fig. EV4B), while it did not improve the neuronal morphology (Fig. EV4C). These data together suggest that, depending on their specific mechanism of action, distinct autophagic inducers may differentially impact RTT neuronal features.

## Trehalose administration ameliorates exploratory and locomotor skills of *Mecp2* KO mice

We investigated whether in vivo administration of trehalose could counteract disease pathology and improve behavioral phenotypes of *Mecp2* KO mice. WT and KO animals were intraperitoneally injected every other day with 2 g/kg of trehalose or with physiological solution from P18 to P40 and subjected to a panel of behavioral assessments (Fig. 7A). Firstly, we evaluated RTT-related phenotypes using the well-established Bird scoring system (mobility, hindlimb clasping, tremor, gait and general condition) (Guy et al, 2007), twice a week until 30 days after treatment ending. Although KO-treated animals did not display an overall amelioration compared to untreated animals (Fig. EV5A), they improved their motor coordination on the pole test (Fig. 7B). While *Mecp2* KO mice manifested an increased descending time when placed on the top of the pole with respect to WT animals, KO-treated mice reached the bottom in a shorter time (Fig. 7B). Consistently with literature data (Guy et al, 2001), KO mice also showed hypoactivity in an open area, traveling less distance and exhibiting slower movement. Trehalose administration resulted in the rescue of total distance traveled (Fig. 7C,D) and movement speed (Fig. EV5B). We also observed a significant improvement in the anxiety-like behavior of *Mecp2* KO-treated mice as they move in the center of the arena similarly to control animals (Fig. 7E). To further validate exploratory and anxiety-like behavior, we extrapolated the rearing activity by counting the number of times a mouse stood on their hind legs and rose forelimbs to lean (supported) or not (unsupported) to the arena's wall. Supported rearing is recognized as a sign of explorative behavior, while unsupported rearing negatively correlates with anxiety features. While untreated *Mecp2* KO animals manifested reduced supported activity counts, treated mice performed as well as WT littermates (Fig. 7F). Similarly, unsupported rearing resulted decreased in mutant animals, although not statistically significant, but again it was normalized by trehalose treatment (Fig. EV5C). Finally, a marble test was used to evaluate whether trehalose treatment could improve the defective environment-directed exploratory behavior of *Mecp2*-KO mice. As shown in Fig. EV5D, compared to WT, mutant mice manifested less ability to bury beads, which was only partially normalized by trehalose treatment. Overall, these results indicate that trehalose administration ameliorated motor and exploratory habits of *Mecp2* KO mice, as well as anxiety-related

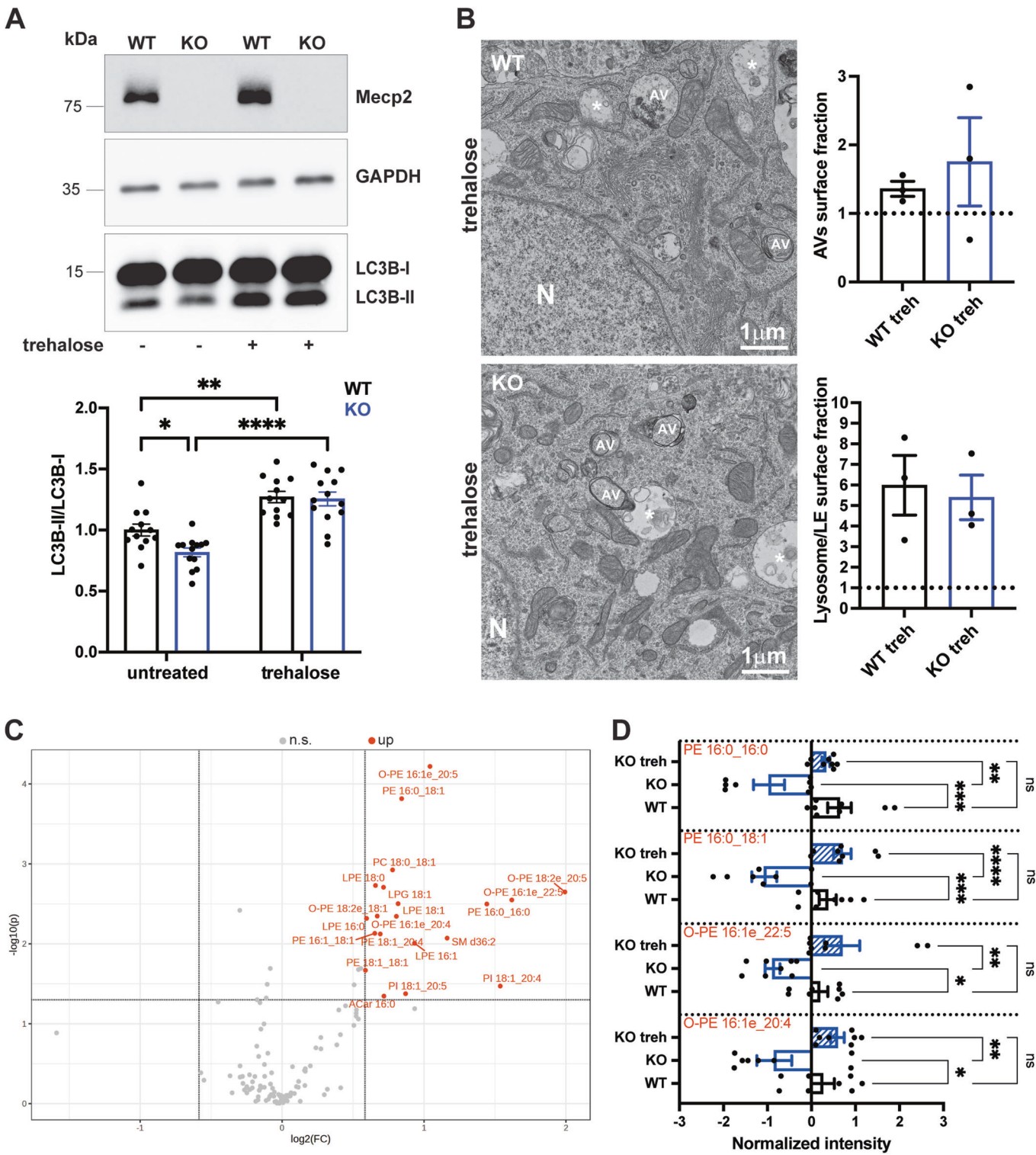

phenotypes. It is worth noting that no significant difference was detected among WT and WT-treated animals in any of the behavioral tests performed, thus confirming the safety of this molecule.

As trehalose has been shown to increase the lifespan in several animal models of neurological disorders (Honda et al, 2010; Li et al, 2015; Palmieri et al, 2017; Tanaka et al, 2004; Zhang et al, 2014), we tested whether it could also ameliorate the life expectancy of *Mecp2* KO mice, which normally lasts about 4 months of age. Although we did not find a striking effect on the survival of Mecp2 KO-treated animals, we could observe that 50% of them lived 15 days longer than untreated ones (Fig. EV5E).

**Figure 4. Trehalose treatment restores the autophagosome biogenesis in *Mecp2* KO neurons.**

(A) Representative western blot from lysates of WT and KO cortical neurons (14 DIV) under resting condition or incubated with 25 mM trehalose for 48 h. LC3B-I and LC3B-II intensities were quantified by densitometric analysis. LC3B-II/LC3B-I ratio was calculated. Mecp2 signal is shown as a genotype control. GAPDH was used as a loading control. Data were expressed as mean ± SEM ($n = 12$–13 embryos from five independent experiments). Two-way ANOVA with Tukey's multiple comparisons test, *$p < 0.05$, **$p < 0.01$, ****$p < 0.0001$ (LC3B-II/I: ut WT vs ut KO $p = 0.0437$, ut WT vs treh WT $p = 0.0015$, ut WT vs treh KO $p = 0.0024$, ut KO vs treh WT $p < 0.0001$, ut KO vs treh KO $p < 0.0001$, treh WT vs treh KO $p = 0.9957$). (B) Representative TEM micrographs of WT and KO cortical neurons (14 DIV) incubated with trehalose for 48 h. The area occupied by autophagosomes and autolysosomes (AVs) or by lysosomes over the total cell area in the image was quantified. Data were expressed as mean ± SEM normalized on WT untreated (data shown in Fig. 1E, dashed line on the graph) ($n = 3$ independent experiments, with 58 WT and 61 KO treh-treated neurons analyzed). N nucleus, AV autophagosome, *=enlarged lysosomes. (C) Volcano plot comparing KO vs KO treated cortical neurons lipidoms. Equal variance (unpaired) two tails $t$-test, a threshold set to $p < 0.05$ and fold change >1.5 (red: lipids upregulated in KO treh; PE: diacylglycerophosphoethanolamines; O-PE: alkyl, acylglycerophosphoethanolamines; PS: diacylglycerophosphoserines; LPG: monoacylglycerophosphoglycerols; SM: ceramide phosphocholines/sphingomyelins; PC: diacylglycerophosphocholines; ACar: acetylcarnitines; LPE: diacylglycerophosphoethanolamines; PI: diacylglycerophosphoinositols). (D) Boxplot of selected lipids are shown. Data were expressed as mean ± SEM ($n = 4$ embryos with two technical replicates, from three independent experiments). One-way ANOVA followed by Tukey's multiple comparison test, *$p < 0.05$, **$p < 0.01$, ***$p < 0.001$, ****$p < 0.0001$ (PE 16:0_16:0: WT vs KO $p = 0.0008$, WT vs KO treh $p = 0.6862$, KO vs KO treh $p = 0.0058$; PE 16:0_18:1: WT vs KO $p = 0.0006$, WT vs KO treh $p = 0.5680$, KO vs KO treh $p < 0.0001$; O-PE 16:1e_22:5: WT vs KO $p = 0.03$, WT vs KO treh $p = 0.4042$, KO vs KO treh $p = 0.0015$; O-PE 16:1e_20:4: WT vs KO $p = 0.0347$, WT vs KO treh $p = 0.6949$, KO vs KO treh $p = 0.0056$). Source data are available online for this figure.

## Discussion

Alterations in the autophagic pathway have been observed in many disorders, including metabolic and neurodegenerative diseases (Kitada and Koya, 2021; Nixon, 2013). However, relatively few studies investigated autophagy dysfunction in neurodevelopmental disorders (Esposito et al, 2019; Fassio et al, 2018, 2020; Guerrini et al, 2022; Le Duc et al, 2019; Wang et al, 2016; Zapata-Muñoz et al, 2021), even if this cellular process regulates crucial aspects of nervous system development including neuroprogenitor proliferation, neuronal maturation, and synaptic formation (Cecconi et al, 2007; Dragich et al, 2016; Stavoe et al, 2016; Tang et al, 2014).

Autophagy signaling has been only partially studied in cellular and animal models of RTT, with controversial results. Ricciardi and colleagues found a decreased AKT-mTOR signaling in brain cortices of *Mecp2* KO symptomatic mice, which should in turn promote autophagic function, even though this was not addressed in the study (Ricciardi et al, 2011). Nott and co-workers demonstrated that neural progenitor cells (NPCs) derived from an RTT patient harboring an *MECP2* mutation impairs the localization of FOXO3 - one of the major transcription factors in autophagy regulation - on its target promoters, thus negatively affecting their gene expression (Nott et al, 2016). More recently, Sbardella et al observed defective autophagosome biogenesis and accumulation of mitochondria in cellular models of the disease (Sbardella et al, 2017). However, none of these studies further explored autophagic functionality, especially in neurons and brains of RTT models.

To shed light on this, we explored the autophagic cascade in *Mecp2* deficient models and found reduced LC3B-II levels in mouse neuronal cultures, as well as in mouse cortices and RTT human fibroblasts. This corresponded to a lower amount of autophagic structures in KO neurons, in the absence of overt lysosomal defects. Deeper metabolic investigation in *Mecp2* KO neurons revealed a defective phospholipid content mainly affecting PE species necessary for autophagosome maturation. We thus hypothesize that autophagosome biogenesis is impaired in the absence of Mecp2, in part due to lipidic imbalance. In agreement with our data, previous studies showed reduced levels of phosphatidylcholine and PE species in the cerebrospinal fluid of RTT patients (Zandl-Lang et al, 2022). Obviously, given the importance of phospholipid composition in general membrane trafficking, including synaptic vesicle recycling and endocytic trafficking

(Haucke and Di Paolo, 2007), such a perturbed lipid composition is not expected to impact only on autophagy maturation in *Mecp2*-depleted cells (Xu and Pozzo-Miller, 2017).

Interestingly, the occurrence of LC3B-II defects in vivo was identified at early postnatal stages, in a pre-symptomatic phase of the disease, thus underlining a possible link between delayed brain maturation (Bedogni et al, 2016) and defective autophagic function. In vitro studies on cultured neurons also confirmed the parallel appearance of autophagy defects with impairment of dendritic complexity, neuronal activity, and synapse organization. Although, at this stage, we cannot prove that reduced autophagosome biogenesis is directly caused by Mecp2 deficiency rather than by indirect mechanisms, we do believe that defective autophagy participates in the manifestation of disease phenotypes. Accordingly, autophagy enhancement mediated by trehalose normalized the neuronal defects typical of RTT. Dendritic complexity and synaptic ultrastructure were restored upon treatment, thus supporting the idea that efficient autophagy sustains neuronal maturation. However, neuronal activity assessed by calcium imaging was only partially restored by trehalose, and no effect was seen on synaptic activity, thus underlining that other molecular mechanisms contribute to the perturbation of these complex functions and that a combination of drugs might be envisaged to fully recover neuronal functionalities.

Transcriptomic analyses revealed that trehalose modulated gene networks related to metabolic pathways that are altered by Mecp2 absence. Trehalose was also able to significantly promote phospholipid synthesis especially in *Mecp2* KO neurons. This implies that probably, in addition to stimulation of autophagy, trehalose treatment modulates other membrane trafficking events that might participate in the final neuronal outcomes that we observed. Indeed, the use of a different autophagic inducer, the mTOR inhibitor rapamycin, resulted in the potentiation of neuronal activity assessed by calcium imaging but not of morphological parameters in KO neurons.

Trehalose was successfully used in vivo in the treatment of several neurodegenerative disorders with impaired autophagy function (Castillo et al, 2013; Palmieri et al, 2017; Tanaka et al, 2004); it is an FDA-approved molecule and has an excellent safety profile (Morales-Carrizales et al, 2023). In our study, we decided to target the early symptomatic stage of the KO mouse model, which generally correlates with first developmental milestones and

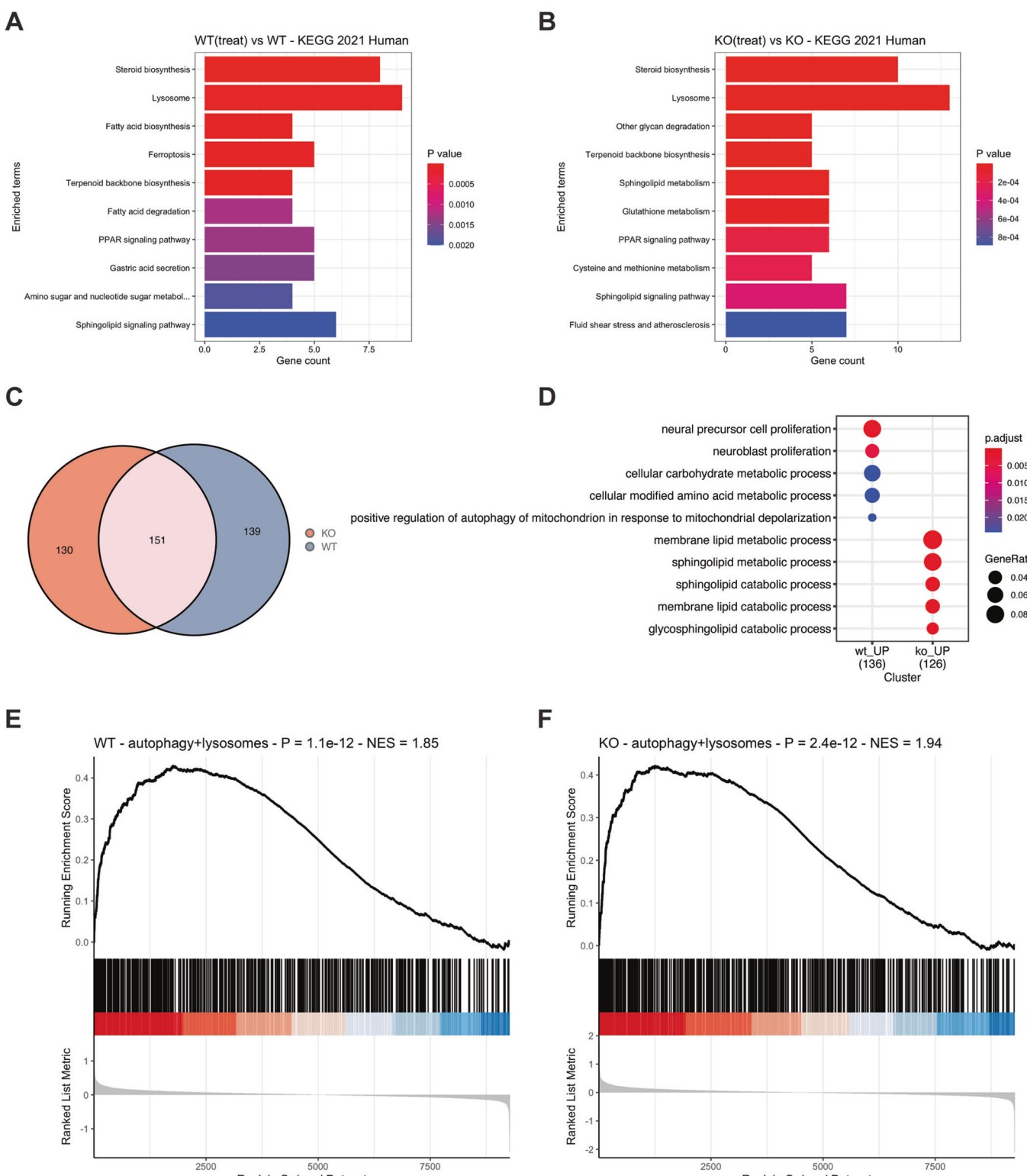

diagnosis occurrence in RTT girls. Importantly, in vivo administration of trehalose in mutant male mice ameliorated locomotor skills, exploratory activity, and anxiety-related behaviors. *Mecp2* KO animals suffered from hypoactivity, motor coordination defects, and tended to remain in the arena periphery nearby the

wall, while treated mice were able to travel the same distance as WT littermates and felt comfortable in moving in central "unprotected" areas. Finally, trehalose delayed the disease progression of *Mecp2* KO-treated mice, but it did not prolong their lifespan, possibly because of the short duration of treatment. We thus hypothesize

**Figure 5. Trehalose- mediated autophagy enhancement upregulates lipid metabolism.**

(A, B) Enriched categories of upregulated genes upon trehalose administration with at least 0.25 log2fold change in expression levels in WT (A) and KO (B) neurons compared to untreated control. The vertical axis shows the top ten KEGG terms while the horizontal axis displays the gene counts belonging to each category. The analysis demonstrated that the majority of enriched processes were related to lipid-metabolic function in both genotypes. The $p$ value was computed using Fisher's exact test. (C) Venn Diagram indicating the number of genes found upregulated in WT, KO, or both. (D) GO of genes upregulated either in WT or KO neurons upon trehalose treatment. The $p$ value was computed using the hypergeometric test. The adjusted $p$ value is obtained using the Benjamini–Hochberg method for correction for multiple hypotheses testing. (E, F) GSEA of transcriptomic changes following trehalose administration in WT (left) and KO (right) neurons related to autophagy-lysosomal genes. The y-axis represents the enrichment score (ES) while the vertical black lines on the x-axis represent the autophagy-lysosomal genes. The colored band at the bottom shows the ranked gene expression of RNA-seq data (left, red: upregulated; right, blue: downregulated) and the degree of correlation with autophagy-lysosomal genes. The significance threshold was set at FDR <0.05 and statistical analysis was computed according to Subramanian et al, 2005. Source data are available online for this figure.

that a long-term intraperitoneal administration in addition to the oral route could result in more robust beneficial effects. In the future, it will be relevant to investigate the therapeutic potential of trehalose also in heterozygous females, which better represent the genetic condition of RTT patients, even though they manifest a milder phenotype and delayed disease onset as compared to KO males. Furthermore, testing the efficacy of trehalose, and possibly other autophagy modulators, in RTT patient induced pluripotent stem cell (iPSC)-derived neurons will pave the way for novel therapeutic strategies.

In conclusion, our findings strengthen the hypothesis that dysfunctions of autophagy and related signaling pathways, such as lipid metabolism, might represent previously neglected molecular mechanisms that contribute to RTT pathogenesis. Therefore, we believe that our data constitute a proof of principle for the investigation of novel therapies to treat devastating neurodevelopmental disorders, such as RTT, based on the concept of autophagy enhancement.

# Methods

### Reagents and tools table

| Reagent/resource | Reference or source | Identifier or catalog number |
|---|---|---|
| **Experimental models** | | |
| Human control fibroblasts | Coriell Institute of Medical Research | GM03651 |
| c.705delG mutation | Coriell Institute of Medical Research | GM07982 |
| B6.129P2(C)-*Mecp2*tm1.1Bird/J | The Jackson Laboratory | JAX:003890 |
| **Recombinant DNA** | | |
| pEGFP-N1-TFEB | Addgene | 38119 |
| pDEST-CMV mCherry-EGFP-LC3B WT | Addgene | 123230 |
| **Antibodies** | | |
| Rabbit anti-LC3B | NovusBio | NB100-2220 |
| Rabbit anti-p62 | Sigma-Aldrich | P0067 |
| Rabbit anti-GAPDH | Cell signaling | 2118 |
| Rabbit anti-p70S6 kinase | Cell signaling | 9202 |
| Rabbit anti-phospho-p70S6 kinase (Thr389) | Cell signaling | 9205 |
| Rabbit anti-ATG5 | Cell signaling | 12994 |

| Reagent/resource | Reference or source | Identifier or catalog number |
|---|---|---|
| Rabbit anti-ATG3 | Cell Signaling | 3415 |
| Rabbit anti-ATG16L | Cell Signaling | 8089 |
| Rabbit anti-LAMP1 | Cell signaling | 3243 |
| Rabbit anti-MeCP2 | Cell signaling | 3456 |
| Rabbit anti-PSD-95 | Invitrogen | 51-6900 |
| Rabbit anti-NeuN | Abcam | ab177487 |
| Rabbit anti-SNAP25 | Abcam | ab5666 |
| Rabbit anti-VAMP2 | Synaptic System | 104211 |
| Secondary HRP-conjugated | Jackson ImmunoResearch | 111-035-144 115-035-003 112-035-003 |
| Rabbit anti-MeCP2 | Cell signaling | 3456 |
| Chicken anti-GFP | Thermo Fisher | A10262 |
| Rabbit anti-MAP2 | Cell Signaling | 8707 |
| Goat anti-chicken Alexa Fluor Plus 488 | Thermo Fisher | A32931 |
| Goat anti-rabbit Alexa Fluor 647 | Immunological Sciences | IS20043 |
| Goat anti-rabbit Alexa Fluor 568 | Thermo Fisher | A11036 |
| **Oligonucleotides and other sequence-based reagents** | | |
| *Mecp2* forward | 5′-AAATTGGG TTACACCGCTGA-3′ | |
| *Mecp2* reverse KO | 5′-CCACCTAGC CTGCCTGTACT-3′ | |
| *Mecp2* reverse WT | 5′-CTGTATCCTTG GGTCAAGCTG-3′ | |
| *Jarid1d/Kdm5d* forward | 5′-CCAGGATCTGAC GACTTTCTACC-3′ | |
| *Jarid1d/Kdm5d* reverse | 5′-TTCTCCGCAA TGGGTCTGATT-3′ | |
| Xtra Taq Pol | Genespin | XSTS-T5XRTL GL |
| Phire Animal Tissue Direct PCR Kit | Thermo Fisher | F140WH |
| **Chemicals, Enzymes and other reagents** | | |
| Hank's Buffered Salt Solution (HBSS) | Thermo Fisher | 14175095 |
| Trypsin/EDTA | Thermo Fisher | 25200056 |
| Neurobasal Plus medium | Thermo Fisher | A3582901 |
| B-27™ Plus | Thermo Fisher | A3582801 |

| Reagent/resource | Reference or source | Identifier or catalog number |
|---|---|---|
| GlutaMAX-I | Thermo Fisher | 35050038 |
| Pen/Strep | Sigma-Aldrich | P0781 |
| Poly-D-lysine hydrobromide | Sigma-Aldrich | P7886 |
| Lipofectamine™ 2000 | Thermo Fisher | 11668019 |
| EMEM | Merck | M5650 |
| Fetal bovine serum (FBS) | Merck | F7524 |
| L-glutamine | Sigma-Aldrich | G7513 |
| Trehalose | Sigma-Aldrich | T9531 |
| Chloroquine diphosphate | Sigma-Aldrich | C6628 |
| Protease inhibitor cocktail | Sigma-Aldrich | P8340 |
| Leupeptin | Sigma-Aldrich | L2884 |
| Phosphate-buffered saline (DPBS) 10X | Thermo Fisher | 14200091 |
| Formaldehyde 16%, methanol-free | Thermo Fisher | 28908 |
| Triton™ X-100 | Sigma-Aldrich | T8787 |
| BSA (bovine serum albumin) | Sigma-Aldrich | A7906 |
| Hoechst-33342 | Invitrogen | H3570 |
| DAPI | Sigma-Aldrich | D8537 |
| Fluoromount mounting media | Sigma-Aldrich | F4680 |
| Glutaraldehyde | Sigma-Aldrich | G5882 |
| Osmium tetroxide | Electron Microscopy Sciences | 19190 |
| Potassium ferrocyanide | Sigma-Aldrich | 60279 |
| Uranyl acetate | Electron Microscopy Sciences | 22400 |
| Epoxy resin | Sigma-Aldrich | 45359 |
| PureZOL | Bio-Rad | 7326890 |
| DNAse | Sigma-Aldrich | AMPD1 |
| ACQUITY UPLC CSH C18 Column 1.7 μm, 2.1 × 100 mm Column | Waters™ | 186005297 |
| Purine and Hexakis biopolymer | Agilent Technologies | G1969-85001 G1969-85000 |
| Fluo-4 | Invitrogen | F14201 |
| Western Sun | Cyanagen | XLS063,0250 |
| Western Antares | Cyanagen | XLS141,0250 |
| Neurobasal Medium, minus phenol red | Thermo Fisher | 12348017 |
| Rapamycin | MedChemExpress | HY-10219 |
| NMDA | TOCRIS | 0114 |
| Trans-Blot Turbo Mini 0.2 μm Nitrocellulose Transfer Packs | Bio-Rad | 1704158 |
| 30% Acrylamide/bis-acrylamide | Bio-Rad | 1610158 |
| 4X Laemmli sample buffer | Bio-Rad | 1610747 |
| Precision Plus Prestained Protein Standard | Bio-Rad | 1610373 |
| 12-mm coverslips PDL | Neuvitro | GG-12-PDL |

| Reagent/resource | Reference or source | Identifier or catalog number |
|---|---|---|
| **Software** | | |
| ImageJ / Fiji | | https://doi.org/10.1038/nmeth.2019 https://doi.org/10.1038/nmeth.2089 |
| Image Lab 6.1 | Bio-Rad | |
| Velox software | FEI, Thermo Fisher Scientific | |
| Microscope Image Browser, MIB | | https://doi.org/10.1371/journal.pbio.1002340 |
| Spectral matching software MassHunter Lipid Annotator | Agilent Technologies, USA | |
| MetaboAnalyst | | https://doi.org/10.3390/metabo10050186 |
| HCS Studio software using SpotDetector bioapplication | Thermo Fisher Scientific | |
| EthoVision 14 | Noldus | |
| Graphpad 7.0 | Graphpad Software Inc., La Jolla, CA | https://www.graphpad.com/ |
| BioRender | BioRender | https://www.biorender.com/ |
| **Other** | | |
| Trans-blot Turbo Transfer system | Bio-Rad | |
| Spinning-disk confocal Nikon Eclipse Ti | Nikon | |
| ORCA-Flash4.0 digital camera C13440 | Hamamatsu | |
| Ultramicrotome | UC7, Leica microsystem | |
| Transmission Electron Microscope Talos L120C | FEI, Thermo Fisher Scientific | |
| Ceta CCD camera | FEI, Thermo Fisher Scientific | |
| 1290 Infinity II LC | Agilent | |
| 6546 LC/Q-TOF system | Agilent | |
| Liquid handling system of the ArrayScan XTI HCA Reader | Thermo Fisher Scientific | |
| T100 TM Thermal Cycler | Bio-Rad | |
| ChemiDoc MP | Bio-Rad | 12003154 |
| Zeiss Axio Observer.Z1with Hamamatsu EM 9100 | Zeiss | |

## Animals

The *Mecp2* KO mouse strain was originally purchased from The Jackson Laboratory (B6.129P2(C)-*Mecp2*^tm1.1Bird^/J, RRID:IMSR_-JAX:003890), then backcrossed and maintained on a CD1 background, as previously described (Cobolli Gigli et al, 2016). Heterozygous females were bred to WT CD1 males to give rise to WT and KO litters. Only male mice were used throughout the

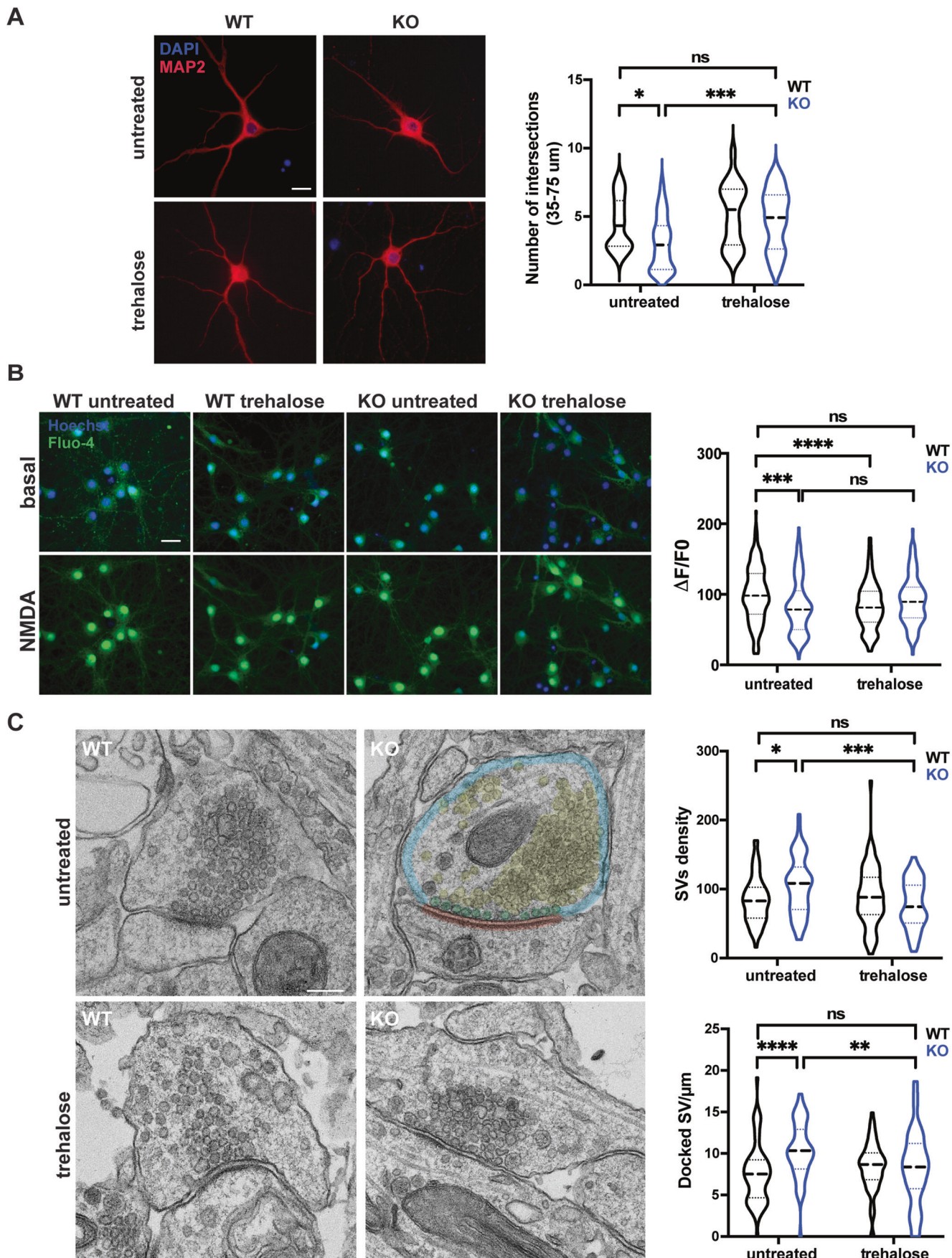

◄ **Figure 6. Trehalose treatment improves morphological and functional alterations observed in *Mecp2* KO neurons.**

(A) Representative images of MAP2 positive WT and KO primary cortical neurons (6 DIV) treated or not with trehalose (50 mM for 48 h) and morphologically analyzed with the Sholl analysis plugin. Scale bar: 20 μm. The number of intersections in the range of 35–75 μm from soma is quantified. Two-way ANOVA followed by Tukey's multiple comparisons test, *$p < 0.05$, ***$p < 0.001$ (ut WT vs ut KO $p = 0.0173$, ut WT vs treh WT $p = 0.4588$, ut WT vs treh KO $p = 0.8563$, ut KO vs treh WT $p < 0.00001$, ut KO vs treh KO $p = 0.0005$, treh WT vs treh KO $p = 0.8879$). (B) Representative images of WT and KO primary cortical neurons (14 DIV) loaded with Fluo-4, before and after exposure to 100 μM NMDA. Scale bar: 40 μm. Data indicate the Fluo-4 intensity (ΔF/F0) of WT and KO treated neurons and their relative controls (violin plots, median ± quartiles; $n = 216$ cells for WT untreated, $n = 123$ cells for KO untreated, $n = 236$ cells for WT treated, $n = 141$ cells for KO treated from three independent experiments). Two-way ANOVA followed by Tukey's multiple comparisons test, ***$p < 0.001$, ****$p < 0.0001$ (ut WT vs ut KO $p = 0.0002$, ut WT vs treh WT $p < 0.0001$, ut WT vs treh KO $p = 0.1083$, ut KO vs treh WT $p = 0.9982$, ut KO vs treh KO $p = 0.2632$, treh WT vs treh KO $p = 0.2152$). (C) Representative TEM images of synaptic terminals from WT and KO cortical neurons (14 DIV) under resting conditions or incubated with 25 mM trehalose for 48 h. The area of the presynaptic terminal (blue), the number of synaptic vesicles (SVs, yellow), the number of docked SVs (green), and the length of the active zone (AZ, red) were quantified. The density of SVs in the terminal and the density of SVs docked to the active zone are shown. Scale bar: 200 nm. Data were expressed as violin plots, median ± quartiles ($n > 60$ cells analyzed/ condition for each experiment, from three independent experiments). Two-way ANOVA followed by Tukey's multiple comparisons tests, *$p < 0.05$, **$p < 0.01$, ***$p < 0.001$, ****$p < 0.0001$ (SVs density: ut WT vs ut KO $p = 0.0184$, ut WT vs treh WT $p = 0.8142$, ut WT vs treh KO $p = 0.8073$, ut KO vs treh WT $p = 0.1577$, ut KO vs treh KO $p = 0.0008$, treh WT vs treh KO $p = 0.2784$; Docked SV/μm: ut WT vs ut KO $p < 0.0001$, ut WT vs treh WT $p = 0.5949$, ut WT vs treh KO $p = 0.7447$, nt KO vs treh WT $p = 0.0073$, ut KO vs treh KO $p = 0.0031$, treh WT vs treh KO $p = 0.9947$). Source data are available online for this figure.

study, both for in vivo and in vitro assays. Mouse genotyping and sex determination on single embryos or pups were performed by conventional PCR (Xtra Taq Pol; Genespin, XSTS-T5XRTL GL) on genomic DNA purified from a tail biopsy (Phire Animal Tissue Direct PCR Kit; Thermo Fisher, F140WH), using the following primers: *Mecp2* forward AAATTGGGTTACACCGCTGA, *Mecp2* reverse KO allele CCACCTAGCCTGCCTGTACT, *Mecp2* reverse WT allele CTGTATCCTTGGGTCAAGCTG (expected band size: WT allele 600 bp, KO allele 400 bp); *Jarid1d/Kdm5d* forward CCAGGATCTGACGACTTTCTACC, *Jarid1d/Kdm5d* reverse TTCTCCGCAATGGGTCTGATT (expected band size: 113 bp only in males). Animals were housed in an SPF facility (IRCCS San Raffaele Scientific Institute, Milan), with 12 h light/dark cycle, food and water *ad libitum*, while an inverted cycle was employed for the behavioral studies at the Mouse Behavior Core Facility at San Raffaele Scientific Institute. All experiments involving animals were performed upon authorization from the Italian Ministry of Health, according to international guidelines for animal welfare (European Directive 2010/63/EU).

## Primary neuronal cultures

Primary cortical neurons were prepared from WT and *Mecp2* KO male embryos at embryonic day 15.5 (E15.5). The pregnant dam was sacrificed, and the embryos were collected and sacrificed by decapitation. A tail biopsy was taken from each embryo to perform genotyping and sex determination. Brains were exposed, meninges removed, and the cerebral cortex from both hemispheres was rapidly dissected and kept in ice-cold Hank's Buffered Salt Solution (HBSS; Thermo Fisher, 14175095), a single embryo per tube. Tissues were washed in HBSS, incubated with 0.25% trypsin/EDTA (Thermo Fisher, 25200056) for 10 min at 37 °C, washed in HBSS and mechanically dissociated in Neurobasal Plus medium (Thermo Fisher, A3582901) supplemented with 2% B-27™ Plus (Thermo Fisher, A3582801), 2 mM GlutaMAX-I (Thermo Fisher, 35050038) and 1% Pen/Strep (Sigma-Aldrich, P0781). Neurons were plated in the same medium on plasticware coated with 0.01 mg/ml poly-ᴅ-lysine hydrobromide (Sigma-Aldrich, P7886), at a density of 300,000 cells/well in six-well plates or 1.5 million cells/dish in 6-cm dishes. For immunofluorescence experiments, neurons were plated on 13-mm or 24-mm poly-ᴅ-lysine-coated (0.1 mg/ml) glass coverslips, at a density of 25,000 or 60,000 cells/coverslip.

## Neuron transfection

For the evaluation of TFEB subcellular localization, neurons were transfected with pEGFP-N1-TFEB plasmid (Addgene, #38119) at 13 days in vitro (DIV), using Lipofectamine™ 2000 (Thermo Fisher, 11668019). After 24 h of transfection, neurons were fixed and stained as follows.

For the autophagic flux experiments, neurons were transfected with the pDEST-CMV mCherry-EGFP-LC3B WT plasmid (Addgene, #123230) at 13–14 days in vitro (DIV) using Lipofectamine™ 2000 (Thermo Fisher, 11668019). After 12–16 h of transfection, neurons were live imaged as described below.

## Human fibroblasts

Human control fibroblasts from a healthy control subject (GM03651) and fibroblasts from an RTT patient carrying the c.705delG mutation (GM07982) were purchased at Coriell Institute of Medical Research. Cells were grown in EMEM (Merck, M5650) supplemented with 15% heat-inactivated FBS (Merck, F7524), 2 mM ʟ-glutamine (Sigma-Aldrich, G7513), 1% Pen/Strep (Sigma-Aldrich, P0781). Cells were tested for mycoplasma contamination and recently authenticated.

## Drug treatments and sample processing

Neurons were treated at 12 DIV with 25 mM trehalose (Sigma-Aldrich, T9531) in the cell medium. After 48 h of treatment, at 14 DIV, neurons were quickly washed in PBS and either lysed in hot SDS lysis buffer (50 mM Tris-HCl pH 8, 10 mM EDTA, 1% SDS supplemented with 1:200 protease inhibitor cocktail [Sigma-Aldrich, P8340]) for western blotting, or scraped in −20 °C-cold methanol for lipidomics, or fixed in the appropriate fixing solution for immunocytochemistry or electron microscopy experiments. Neurons treated with 100 nM rapamycin (MedChemExpress, HY-10219) in the cell medium were processed as above. Neurons were treated at 13 DIV with 20 μM Leupeptin (Sigma-Aldrich, L2884) in a cell medium. After 24 h of treatment, at 14 DIV, neurons were lysed in a hot SDS lysis buffer. Human fibroblasts at 70–80% confluency were treated with 25 mM trehalose for 48 h, and cells were lysed in hot SDS lysis buffer. In vivo treatment was performed by intraperitoneal injection of 2 g/kg of trehalose (Sigma-Aldrich, T9531) resuspended in sterile saline solution, every other day from

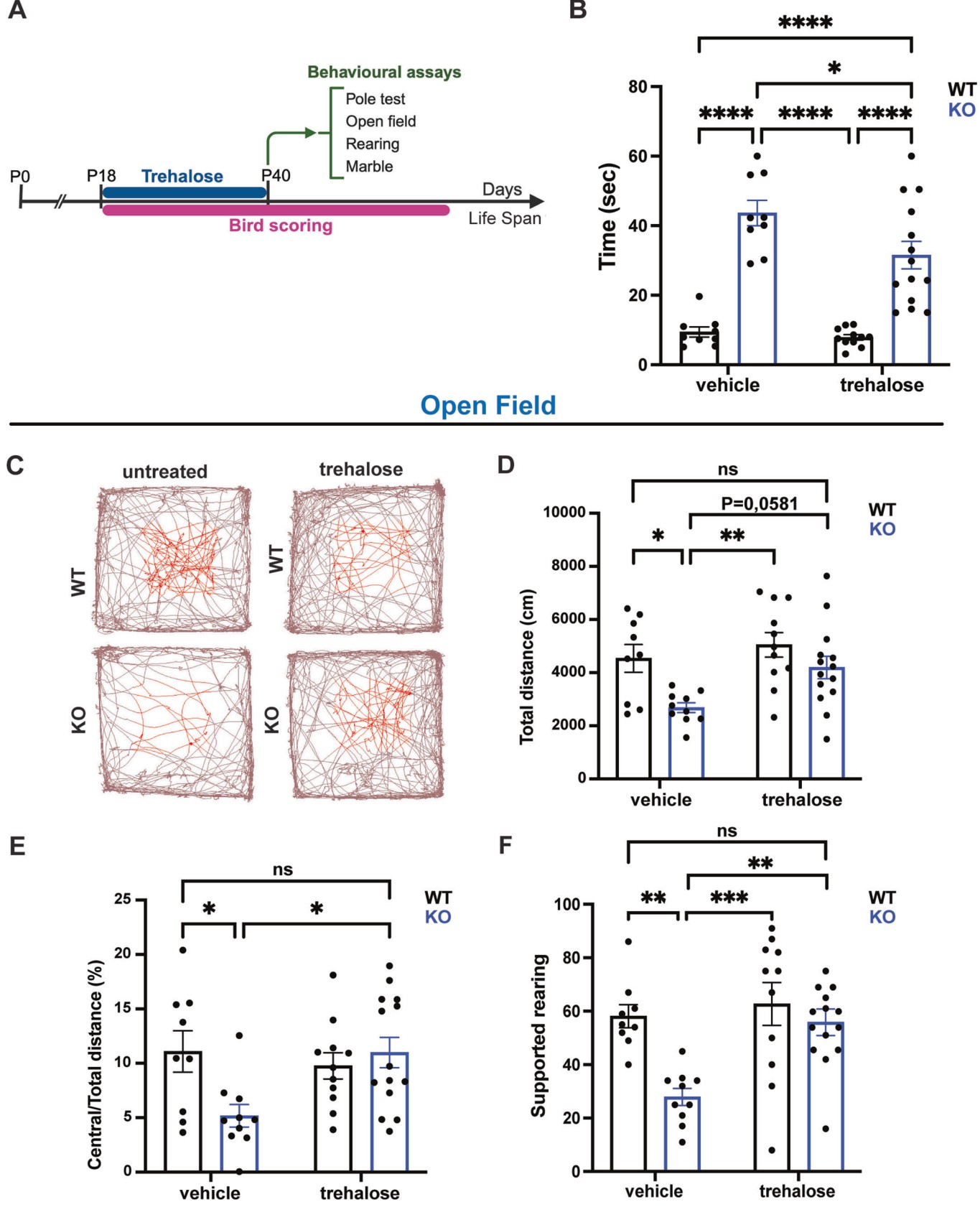

**Figure 7. Trehalose administration ameliorates locomotor and exploratory ability of *Mecp2* KO mice.**

(A) Experimental timeline and outcome measurements. (B) Pole test assay measuring the movement coordination. (C–E) Open field assessment of locomotion and exploratory activity. (C) Mice traces in the arena. (D) Total distance and (E) central/total distance measurements. (F) Supported rearing evaluation. Bars represent mean values with ± SEM. For behavioral analysis $n = 9$ WT vehicle, $10 =$ KO vehicle (9 = KO vehicle in Pole test), $11 =$ WT trehalose, $14 =$ KO trehalose. Behavioral data were analyzed using two-way ANOVA followed by Tukey's multiple comparisons test *$p < 0.05$, **$p < 0.01$, ***$p < 0.001$, ****$p < 0.0001$ (Pole test: ut WT vs ut KO $p < 0.0001$, ut WT vs treh WT $p = 0.9860$, ut WT vs treh KO $p < 0.0001$, ut KO vs treh WT $p < 0.0001$, ut KO vs treh KO $p = 0.0390$, treh WT vs treh KO $p < 0.0001$; Total distance: ut WT vs ut KO $p = 0.0308$, ut WT vs treh WT $p = 0.8503$, ut WT vs treh KO $p = 0.9415$, ut KO vs treh WT $p = 0.0022$, ut KO vs treh KO $p = 0.0581$, treh WT vs treh KO $p = 0.4460$; Central/Total distance: ut WT vs ut KO $p = 0.0417$, ut WT vs treh WT $p = 0.9226$, ut WT vs treh KO $p > 0.9999$, ut KO vs treh WT $p = 0.1262$, ut KO vs treh KO $p = 0.0226$, treh WT vs treh KO $p = 0.9165$; Supported rearing: ut WT vs ut KO $p = 0.0044$, ut WT vs treh WT $p = 0.9407$, ut WT vs treh KO $p = 0.9920$, ut KO vs treh WT $p = 0.0005$, ut KO vs treh KO $p = 0.0058$, treh WT vs treh KO $p = 0.8062$). Source data are available online for this figure.

postnatal day P18 to P40. Control mice were injected with the same volume of sterile saline solution. At the end of the treatment, mice were sacrificed. Cortices were collected, frozen in dry ice, and stored at $-80\,°C$ for subsequent lysis in SDS lysis buffer.

## Nutrient starvation and refeeding

Neurons were starved for 4 h at $37\,°C$ by replacing regular culturing media with a starvation medium (1.8 mM $CaCl_2$, 0.8 mM $MgSO_4$, 5.3 mM KCl, 26.2 mM $NaHCO_3$, 117.2 mM NaCl, 1 mM $NaH_2PO_4$, 5.5 mM D-glucose). Recovery was achieved by replacing the starvation medium with a regular culturing medium followed by 1 h incubation at $37\,°C$. At the end of the procedure, neurons were lysed in a hot SDS lysis buffer as described above.

## Western blotting

Protein lysates were separated by SDS-PAGE. Briefly, 5 μg of protein samples were loaded on 10% or 15% acrylamide gels, run in 25 mM Tris, 192 mM glycine, 0.1% SDS running buffer at 80 V. Protein transfer on nitrocellulose membrane was performed with a semidry apparatus (Trans-blot Turbo Transfer system, Bio-Rad) according to manufacturer's instructions, and assayed by immunoblotting with the following primary antibodies: anti-LC3B (1:1000, rabbit; NovusBio, NB100-2220), anti-p62 (1:1000, rabbit; Sigma-Aldrich, P0067), anti-GAPDH (1:10000, rabbit; Cell signaling, 2118), anti-p70S6 kinase (1:1000, rabbit; Cell signaling, 9202), anti-phospho-p70S6 kinase Thr389 (1:1000, rabbit; Cell signaling, 9205), anti-ATG5 (1:1000, rabbit; Cell signaling, 12994), anti-ATG3 (1:1000, rabbit; Cell Signaling, 3415), anti-ATG16L (1:1000, rabbit; Cell Signaling, 8089), anti-LAMP1 (1:1000, rabbit; Cell signaling, 3243), anti-MeCP2 (1:1000, rabbit; Cell signaling, 3456) anti-PSD-95 (1:500, rabbit; Invitrogen, 51-6900), anti-NeuN (1:5000, rabbit, Abcam, ab177487), anti-SNAP25 (1:5000, rabbit, Abcam, ab5666), anti-VAMP2 (1:10000, mouse, Synaptic System, 104211). Blocking, primary and secondary HRP-conjugated (Jackson ImmunoResearch) antibody incubations were performed in 5% milk or 5% BSA in TBST (150 mM NaCl, 20 mM Tris-HCl pH 7.6, 0.1% Tween-20). ECL detection was performed with Chemidoc MP (Bio-Rad), using Western Sun (Cyanagen XLS063,0250) or Western Antares (Cyanagen XLS141,0250) substrates.

## Immunofluorescence

Neurons were quickly washed with phosphate-buffered saline (PBS, Thermo Fisher) and fixed with pre-warmed 4% paraformaldehyde (Thermo Fisher, 28908) and 10% sucrose in PBS, for 8 min at RT.

Neurons were washed three times with PBS for 10 min at RT, permeabilized with 0.2% Triton™ X-100 (Sigma-Aldrich, T8787) in PBS for 3 min on ice, washed three times in 0.2% BSA in PBS for 10 min at RT. Blocking was performed with 4% BSA in PBS for 15 min at RT. Primary antibodies were diluted in 0.2% BSA in PBS and incubated o.n. at $4\,°C$, followed by three washes in 0.2% BSA in PBS for 10 min at RT. Secondary antibodies were incubated in 0.2% BSA in PBS for 1 h at RT, followed by two washes in 0.2% BSA in PBS for 10 min at RT and two additional washes in PBS for 10 min at RT. Nuclei were stained with Hoechst-33342 (1:10.000 in PBS, 5 min incubation). Coverslips were mounted with Fluoromount mounting media (Sigma-Aldrich, F4680). Primary antibodies used: anti-GFP (1:200, chicken; Thermo Fisher, A10262); anti-MAP2 (1:2000, rabbit; Cell Signaling, 8707). Secondary antibodies used: Goat anti-chicken Alexa Fluor Plus 488 (1:500; Thermo Fisher, A32931); Goat anti-rabbit Alexa Fluor 647 (1:500; Immunological Sciences, IS20043).

Imaging was performed on a spinning-disk confocal Nikon Eclipse Ti, equipped with an ORCA-Flash4.0 digital camera C13440 Hamamatsu and 100X objective. Settings were kept constant for the acquisition of all the images in each experiment.

## Sholl analysis

Cortical neurons at 7 DIV were fixed with 4% paraformaldehyde and 10% sucrose in PBS for 8 min and permeabilized in 0.2% Triton X-100 in PBS for 3 min on ice. Blocking solution (4% BSA in PBS) was added for 15 min at room temperature, followed by incubation with primary antibody, anti-MAP2 (1:500, rabbit; Cell signaling, 8707) o.n. at $4\,°C$. Cells were washed three times in 0.2% BSA—PBS at room temperature, and the secondary antibody Goat anti-rabbit 568 (1:500, Thermo Fisher, A11036) was added for 1 h at RT. Nuclei were stained with DAPI (1:10 in PBS; Sigma-Aldrich, D8537)) for 10 min. Coverslips were mounted with Fluoromount mounting media (Sigma-Aldrich, F4680). Isolated MAP2-positive neurons were imaged with a Nikon epifluorescence microscope equipped with a 40X objective. Dendritic processes were manually traced by using Adobe Photoshop (2017). Traced neurons were binarized by using ImageJ software (2017), and the Sholl analysis plugin was run to investigate the complexity of neuronal arbors. Dendritic intersection numbers were calculated at different discrete distances from the soma, defined by concentric circles spaced by 15 μm and centered on cell nuclei. The total neuronal length is also represented on the x-axis.

## mCherry-EGFP-LC3 assay

For the analysis of autophagic flux with the mCherry-EGFP-LC3 construct, transfected neurons were live-imaged in complete

Neurobasal Medium minus phenol red (Thermo Fisher, 12348017) with controlled temperature and $CO_2$ on an inverted Nikon Eclipse Ti microscope equipped with Yokogawa *CSU-X1* spinning-disk confocal unit, an ORCA-Flash4.0 digital camera C13440 Hamamatsu and 60X objective. Acquisition settings (laser power and exposure time) were differentially adjusted for every image due to high variability in the expression levels of the construct in each cell. When specified, neurons were treated for 2 h at 37 °C with 50 μM chloroquine (Sigma-Aldrich, C6628). The analysis was performed in blind, using the NIS Elements AR software (Nikon). Maximum intensity projection images were produced, a ROI was automatically selected on the cell soma and manually adjusted if needed. A threshold was applied on the green and red channels to allow object autodetection. The number of red, green, and yellow (intersection of red and green) was determined with the Object count tool.

## TFEB-EGFP localization analysis

For TFEB-EGFP localization analysis, neurons were labeled with anti-GFP, anti-MAP2 antibodies and Hoechst as described above. Transfected neurons were manually categorized into cells presenting either: only nuclear localization of TFEB-EGFP; both nuclear and cytoplasmic localization; only cytoplasmic localization. Results were expressed as percentage of neurons belonging to these three categories over the total number of neurons analyzed.

## TEM

Cultured neurons were fixed with 2.5% glutaraldehyde (Sigma, G5882) in 0.1 M cacodylate (Sigma, 20840) buffer pH 7.4 for 1 h at room temperature. Sample were then post-fixed in 1% osmium tetroxide (Electron Microscopy Sciences, 19190), 1.5% potassium ferrocyanide (Sigma, 60279) in 0.1 M cacodylate buffer 1 h on ice and en-bloc stained in 0.5% uranyl acetate (Electron Microscopy Sciences, 22400) overnight at 4 °C. Samples were then dehydrated in increasing concentration of ethanol and infiltrated in epoxy resin (Sigma, 45359). After curing at 60 °C for 48 h embedded cells were removed from the glass coverslips by dipping in liquid nitrogen. Ultrathin sections were obtained using an ultramicrotome (UC7, Leica microsystem, Vienna, Austria), collected copper grids, stained with uranyl acetate and Sato's lead solutions, and observed in a Transmission Electron Microscope Talos L120C (FEI, Thermo Fisher Scientific) operating at 120 kV. Images were acquired with a Ceta CCD camera using Velox software (FEI, Thermo Fisher Scientific). For morphological analysis, images were analyzed by the stereology plugin of Microscope Image Browser, MIB (Belevich et al, 2016). Degradative organelles were classified into two categories: Autophagic vacuoles (AVs) if they display double or single membrane-bound vacuoles that contain portion or remains of cell components (i.e., endoplasmic reticulum, mitochondria, or ribosomes); lysosomes/late endosomes (lys/LE) for single membrane-bound organelles that are heterogeneous in their content with electro-lucent or electron-dense lumen containing small vesicles or membrane whorls (Barral et al, 2022). Synaptic profile area, synaptic vesicle (SV) number, and the length of the active zone (AZ) were determined using ImageJ. SVs were identified as small (~40 nm) round organelles of clear content. The density of SVs in the terminal was quantified as the number of SVs normalized on the synaptic area (μm²), while the density of docked SVs was quantified as the number of SVs adjacent to the AZ membrane normalized on the AZ length (μm).

## Total RNA extraction

After a rapid wash in PBS (Sigma D8537) to remove cellular debris, total RNA was extracted from primary cortical neurons using PureZOL (Bio-Rad 7326890). In details, 0.3 mL of PureZOL (12 wells plate) was added on cell culture and left 5 min at room temperature to allow the complete dissociation of nucleoprotein complexes. About 200 μL of chloroform was then added to 1 ml of PureZOL according to the manufacturer's protocol. Following, samples were inverted for 15–20 s to gently mix the phenol:chloroform mixture and incubated for 3 min at RT. After centrifugation at $12,000 \times g$ for 15 min at 4 °C, the upper aqueous phase was transferred into RNAse-free tube and precipitated with 10 ug of RNA-grade glycogen and 100% isopropanol 1:2. Samples were stored o.n. at −20 °C. The day after, the RNA was centrifuged at $12,000 \times g$ for 10 min at 4 °C and pellets were then washed in 500 μl of 70% ethanol ($7500 \times g$ for 5 min at 4 °C). To remove genomic DNA, sample pellets were treated with DNAse (Sigma-Aldrich, AMPD1) at 37 °C for 15 min. A second purification with PureZOL was executed on RNAs, and their pellets were finally resuspended in 10 μl of RNAse-free $H_2O$.

## RNA-seq and bioinformatic analysis

Total RNA was extracted from eight WT and eight KO cortical neurons (14 DIV) treated with trehalose and eight WT and eight KO untreated controls, as described above, for a total of 32 samples. Only high-quality RNA with an RNA integrity number (RIN) of 8 or higher was used. Library preparation and sequencing have been performed with TruSeq RNA Library Preparation Kit v2 (Illumina) by Genewiz. FASTQ sequencing reads were adapter-trimmed and quality-filtered with Trimmomatic (Bolger et al, 2014), prior to mapping to the mm10 mouse reference genome (https://www.gencodegenes.org/mouse/) with STAR (Dobin and Gingeras, 2016). Gene counts were obtained using featureCounts (Liao et al, 2014). Normalization and differential gene expression analysis (DEG) have been performed with DESeq2 (Love et al, 2014). KO vs. WT DGE analysis was adjusted for cell culture and evaluated both in treated and untreated samples ("~cell_culture + genotype*treatment"). The treatment effect was evaluated independently for WT and KO samples and adjusted for the donor mouse ("mouse_id + treatment"). One WT untreated and corresponding treated sample and two KO untreated and related treated samples were found outliers and removed from the final analyses.

## MS-based lipidomics

Cells were scraped using 500 μl of cold methanol. Then samples were passed to vortex for 30 s, centrifuged at $15,000 \times g$ for 1 min at 0 °C, and the supernatant was separated from the pellet and stored. Following, 250 μl of cold methanol were added to the pellet, vortex for 30 s, and kept on ice for 20 min. This step was repeated twice on the pellet. Samples were then centrifuged at $15,000 \times g$ for 1 min at 0 °C, and the supernatant was separated from the pellet. The supernatants recovered from each step were transferred to a new Eppendorf tube and vacuum-dried. Dried extracts were resuspended with 250 μl of methanol prior to the LC-MS analysis. A quality control pool sample was prepared by mixing a small aliquot (10 uL) from each lipid extract,

to evaluate the stability of the LC-MS system. Features with a relative standard deviation (RSD%) higher than 30 in pooled QC were removed and not considered for further analysis.

The LC-MS analysis was performed using an Agilent 1290 Infinity II LC coupled to a 6546 LC/Q-TOF system. Lipid separation was carried out with an ACQUITY UPLC CSH C18 Column 1.7 μm, $2.1 \times 100$ mm Column (Waters™). Mobile phase A was 10 mM ammonium acetate: acetonitrile (40:60 v:v) with 0.1% formic acid and B was Isopropanol: Phase A (90:10 v:v). A 20-min gradient followed by 1 min of equilibration time was performed as follows: 0 min 99% A, 1 min 99% A, 1.10 min 60% A, 5 min 20% A, 11 min 20% A, 12 min 1% A, 18 min 1% A, 18.10 min 60% A, 20 min 99%. The flow rate was 0.25 ml/min and the column temperature was 55 °C. The resolving power of the mass spectrometer was 50,000, and it operates in a full scan range of m/z 100–1350. The pooled samples were analyzed each five injections and used as quality controls. Finally, four pool injections were analysed in data-dependent acquisition with a fixed collision energy of 30 eV in iterative mode. The samples were analyzed in full scan both in positive and in negative ionization mode, and the injection volume was 2 and 5 μl for positive and negative analysis, respectively. The instrument was calibrated using a TuningMix MMI-L Low concentration (Agilent Technologies, USA). Purine and Hexakis biopolymer (Agilent Technologies, USA) were used as reference mass both in positive and negative experiments and continuously infused at a flow rate of 0.08 ml/min. Lipids annotation was performed on the MS/MS data acquired in DDA mode, using the spectral matching software MassHunter Lipid Annotator (Agilent Technologies, USA) and our in-house database. The annotation parameters were ±5ppm for the mass score, an isotope cluster score higher than 80%, and an MS/MS score higher or equal to 90%. The annotated lipids were used to build an internal lipids library that was then used to annotate and integrate the analytes in the full scan analysis. Data analysis were performed using MetaboAnalyst (https://doi.org/10.3390/metabo10050186). Two different MS-based lipidomics experiments performed at different times were normalized for batch effect using the EigenMS method. After batch correction, missing values were replaced by 1/5 of the minimum positive value of each variable (0.7% of missing values). The dataset was then normalized by the sum of the features, square root transformed and scaled by unit variance scaling method before applying univariate and multivariate analysis. Volcano plots were obtained using a fold change threshold of 1.5 and a $p$ value threshold of 0.05.

## Calcium imaging

Primary cortical neurons were loaded with 2 μM Fluo-4 (Invitrogen) in KRH (Krebs'–Ringer's–HEPES containing (in mM): 125 NaCl; 5 KCl; 1.2 $MgSO_4$; 1.2 $KH_2PO_4$; 25 HEPES; 6 glucose; 2 $CaCl_2$; pH 7.4) for 30 min at 37 °C and then washed once with the same solution. Stimulation was performed automatically by using the liquid handling system of the ArrayScan XTI HCA Reader (Thermo Fisher Scientific). For stimulation, one dose of NMDA (100 μM at the rate of 50 μl/s) was added while images were digitally acquired with a high-resolution camera (Photometrics) through a 20X objective (Zeiss; Plan-NEOFLUAR 0.4 NA). Hoechst fluorescence was imaged as well. Forty frames were acquired at 1 Hz with 40 ms exposure time for Fluo-4 and 25 ms exposure time for Hoechst. At least nine baseline images were acquired before stimulation. The analysis was done with HCS Studio software using

SpotDetector bioapplication (Thermo Fisher Scientific). Hoechst-positive nuclei were identified and counted, and the mean intensity of the Fluo-4 signal was measured in the cell body area of each cell; background intensity was measured and subtracted from the mean intensity. Only cells with neuronal morphology were included in the analysis. Calcium responses were measured as $\Delta F/F_0$.

## Electrophysiology, intracellular recordings

Whole-cell patch-clamp recordings were performed at room temperature from DIV 14 primary mouse cortical neurons, as previously described (Heise et al, 2017). Briefly, primary cortical cultures were perfused with an external solution containing (in mM): 130 NaCl, 2.5 KCl, 2 $CaCl_2$, 1 $MgCl_2$, 10 D-glucose, 10 HEPES-NaOH (pH 7.4 with NaOH). The composition of the intracellular solution was (in mM): 126 K-gluconate, 4 NaCl, 1 EGTA, 1 $MgSO_4$, 0.5 $CaCl_2$, 3 ATP (magnesium salt), 0.1 GTP (sodium salt), 10 glucose, 10 HEPES-KOH. mEPSCs were pharmacologically isolated, supplementing the external solution with TTX (1 μM) and bicuculline (20 μM) to block voltage-gated sodium channels and GABAA receptors, respectively. Currents were acquired in voltage-clamp configuration at a holding potential of -65 mV, filtered at 1 kHz, and digitized at 20 kHz using Clampex 10.1 software. Analysis was performed offline with Clampfit 10.1 software using a threshold-crossing principle.

## Bird scoring

*Mecp2* KO and WT male mice, treated and untreated, were tested according to the scoring system of Guy et al (Jacky Guy et al, 2007). Twice a week, the mobility, hindlimb clasping, tremor, gait, and general conditions were assessed by a blinded operator. Pathological phenotypes were given a score following this scale: 0 if the symptom was absent, 1 if the symptom was present but mild, 2 when the symptom was severe. We also introduced intermediate scores of 0.5 and 1.5 in order to better characterize small variations.

## Open field

The test is performed in an open arena ($50 \times 50$ cm). During the open field test mice were allowed to explore the arena for 10 min. Total distance (cm), central distance (cm), mean speed (cm/s), and rearing activity was measured and represented. Trials were videorecorded and automatically analyzed using the EthoVision 14 (Noldus) software.

## Pole test

Mice were placed head upon the top of a pole and the time taken to orient the body downward and descend on the bottom of the pole was recorded. The test was repeated three times. Maximum time allowed for each trial was 60 s. Mean value of the three trials was calculated.

## Marble assay

Mice were placed individually in a rat cage that was filled with moderately fine wood chip bedding. The bedding was layered up to 10 cm from the cage floor. Twelve glass marbles (diameter: 1.5 cm) were placed uniformly throughout the cage. Mice were allowed to freely explore the cage for 30 min. The number of buried marbles

The paper explained

**Problem**

Autophagy is a highly conserved catabolic process that directs dysfunctional proteins, lipids, and organelles to lysosomes for degradation. Due to their post-mitotic nature, neurons critically rely on autophagy to maintain their homeostasis. While dysfunctional autophagy has been extensively studied in neurodegenerative conditions, its role in neurodevelopmental diseases is still neglected.

Loss of function mutations in the X-linked *MECP2* gene are associated with Rett syndrome, a devastating neurodevelopmental disorder. In addition to severe neurological symptoms, Rett patients and mouse models suffer from a complex metabolic syndrome, suggesting that pathways related to cell metabolism including autophagy might be affected and contribute to disease pathogenesis.

**Results**

We demonstrate that the absence of Mecp2 is associated with impaired autophagosome maturation, probably due to phospholipid imbalances in Mecp2-deficient neurons. Autophagy enhancement with trehalose rebalanced lipid content, autophagic defects, and neuronal features in Rett neurons. In vivo, trehalose treatment ameliorated locomotor and exploratory habits of a Mecp2 knock-out mouse model.

**Impact**

By gaining comprehension of defective autophagic cascade in Mecp2-deficient models, we pose novel therapeutic perspectives for Rett syndrome and for other neurodevelopmental disorders based on the concept of autophagy modulation.

was recorded at the end of the test. A marble was considered "buried" when at least 2/3 of its surface was covered by a litter cage.

## Statistical analysis

The normal distribution of experimental data was assessed using the D'Agostino-Pearson normality test or the Shapiro–Wilk test for TEM data. Data with normal distribution were analyzed by unpaired *t*-test with Welch's correction; non-normally distributed data were analyzed by Mann–Whitney test. Data including four groups (two genotypes and two treatments) were analyzed with two-way ANOVA, followed by Tukey's multiple comparison test. Box plots (Figs. 1A, 2B, C, 3A and EV1) extends from the 25th to the 75th percentiles and include a line at the median value, whiskers indicate the minimum and the maximum values, with all individual data points plotted. All tests were performed with Graphpad 7.0 (Graphpad Software Inc., La Jolla, CA).

## Data availability

The datasets produced in this study are available in the following databases: RNA-Seq data: Gene Expression Omnibus # GSE271893 (https://www.ncbi.nlm.nih.gov/geo/query/acc.cgi?acc=GSE271893). 

The source data of this paper are collected in the following database record: biostudies:S-SCDT-10_1038-S44321-024-00151-w.

## Peer review information

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

## Acknowledgements

This work was supported by the Italy Ministry of Health grant n. GR2019-12371236 to MP, FCG, and J-MC. We are grateful to the Italian parents' association "ProRETT Ricerca" for the inspiration. We thank Dr Patrizia D'Adamo and the Mouse Behavior Core Facility at San Raffaele Scientific Institute for behavioral tests. We thank Dr. Andrea Raimondi, Dr. Elena Vizzoli, and Dr. Desirée Zambroni of the Alembic imaging facility at the San Raffaele Scientific Institute for the technical support of TEM and calcium imaging experiments. The support of Marie Curie H2020 grant n. 845992 to MP for preliminary experimental data is also acknowledged.

## Author contributions

**Alessandro Esposito**: Conceptualization; Formal analysis; Validation; Investigation; Visualization; Methodology; Writing—original draft; Writing—review and editing. **Tommaso Seri**: Formal analysis; Investigation. **Martina Breccia**: Formal analysis; Investigation. **Marzia Indrigo**: Formal analysis; Investigation. **Giuseppina De Rocco**: Formal analysis; Investigation. **Francesca Nuzzolillo**: Formal analysis; Investigation. **Vanna Denti**: Formal analysis; Investigation. **Francesca Pappacena**: Formal analysis; Investigation. **Gaia Tartaglione**: Formal analysis; Investigation. **Simone Serrao**: Formal analysis; Investigation. **Giuseppe Paglia**: Formal analysis; Supervision; Validation; Visualization. **Luca Murru**: Formal analysis; Investigation. **Stefano de Pretis**: Data curation; Formal analysis. **Jean-Michel Cioni**: Supervision; Funding acquisition; Writing—review and editing. **Nicoletta Landsberger**: Resources; Writing—review and editing. **Fabrizia, Claudia Guarnieri**: Conceptualization; Formal analysis; Supervision; Funding acquisition; Visualization; Methodology; Writing—original draft; Writing—review and editing. **Michela Palmieri**: Conceptualization; Formal analysis; Supervision; Funding acquisition; Investigation; Visualization; Methodology; Writing—original draft; Project administration; Writing—review and editing.

Source data underlying figure panels in this paper may have individual authorship assigned. Where available, figure panel/source data authorship is

listed in the following database record: biostudies:S-SCDT-10_1038-S44321-024-00151-w.

## Disclosure and competing interests statement

The authors declare no competing interests.

# Expanded View Figures

**Figure EV1.  Mecp2 deficiency leads to a defective autophagosome maturation.**

(A) Representative western blot from lysates of WT and KO cortical neurons at different days in culture (DIV). LC3B-I and LC3B-II intensities were quantified by densitometric analysis. LC3B-II/LC3B-I ratio was calculated. Mecp2 signal is shown as a genotype control. Data were expressed as median ± min/max ($n = 5$ embryos from three independent experiments). Mann–Whitney test, *$p < 0.05$ (3 DIV $p = 0.8413$; 7 DIV $p > 0.99$; 14 DIV $p = 0.0159$; 21 DIV $p = 0.0556$). (B) Western blot from lysates of WT and KO cortical neurons at different days in culture (DIV). NeuN, PSD-95, SNAP25, and VAMP2 signals are shown as markers of developmental progression ($n = 1$). (C) Representative TEM micrographs of WT and KO cortical neurons (14 DIV) under resting conditions showing examples of autophagic vacuoles in WT neurons, and the presence of immature autophagic structures in neuronal processes in KO neurons. (D) Representative western blot from lysates of WT and KO cortical neurons (14 DIV). VGLUT1 intensity was quantified by densitometric analysis and normalized on GAPDH intensity. Data were expressed as median ± min/max, normalized on WT ($n = 6$ embryos from three independent experiments). Mann–Whitney test, **$p < 0.01$ ($p = 0.0022$).

▶

**A**

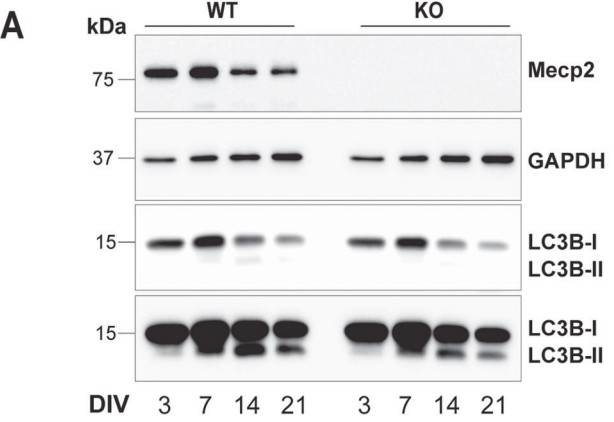

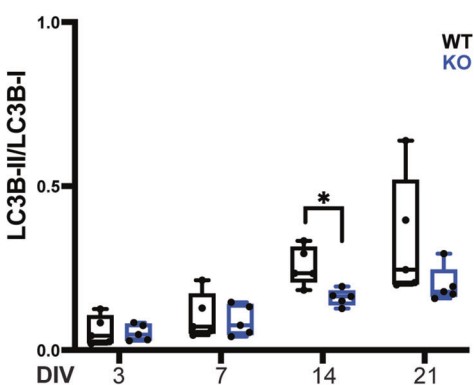

**B**

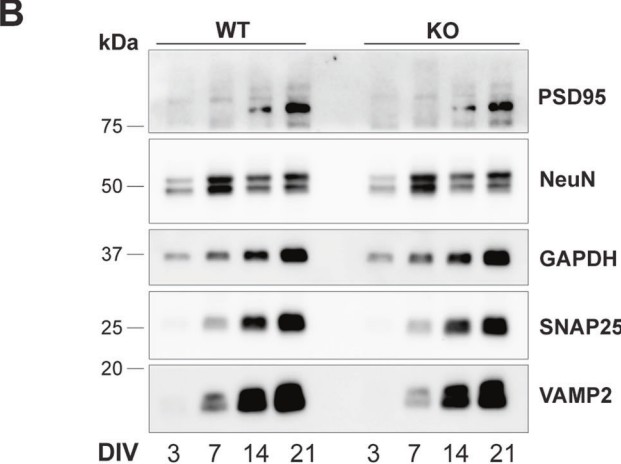

**C**

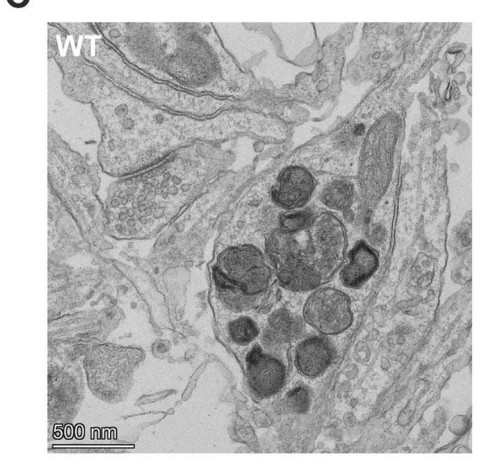

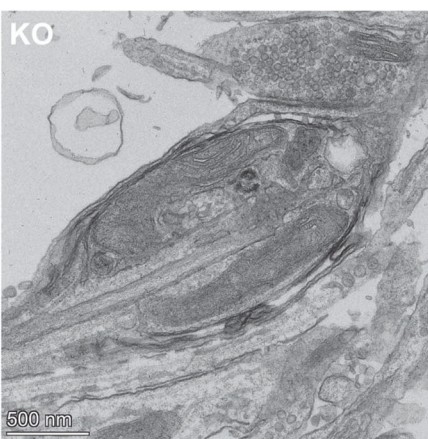

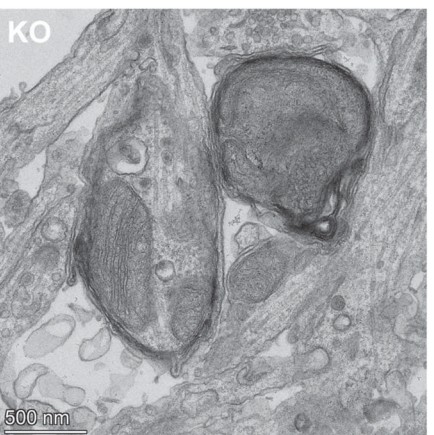

**D**

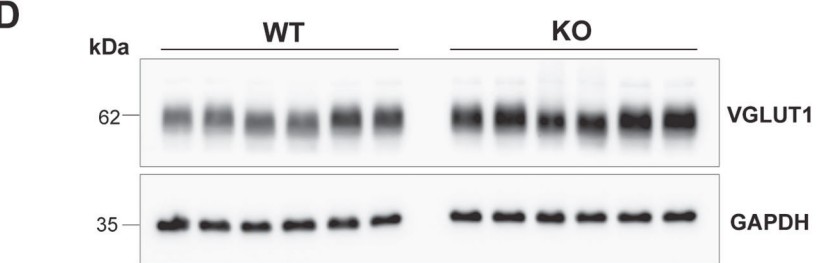

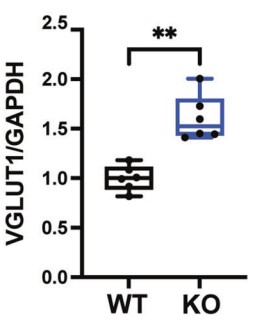

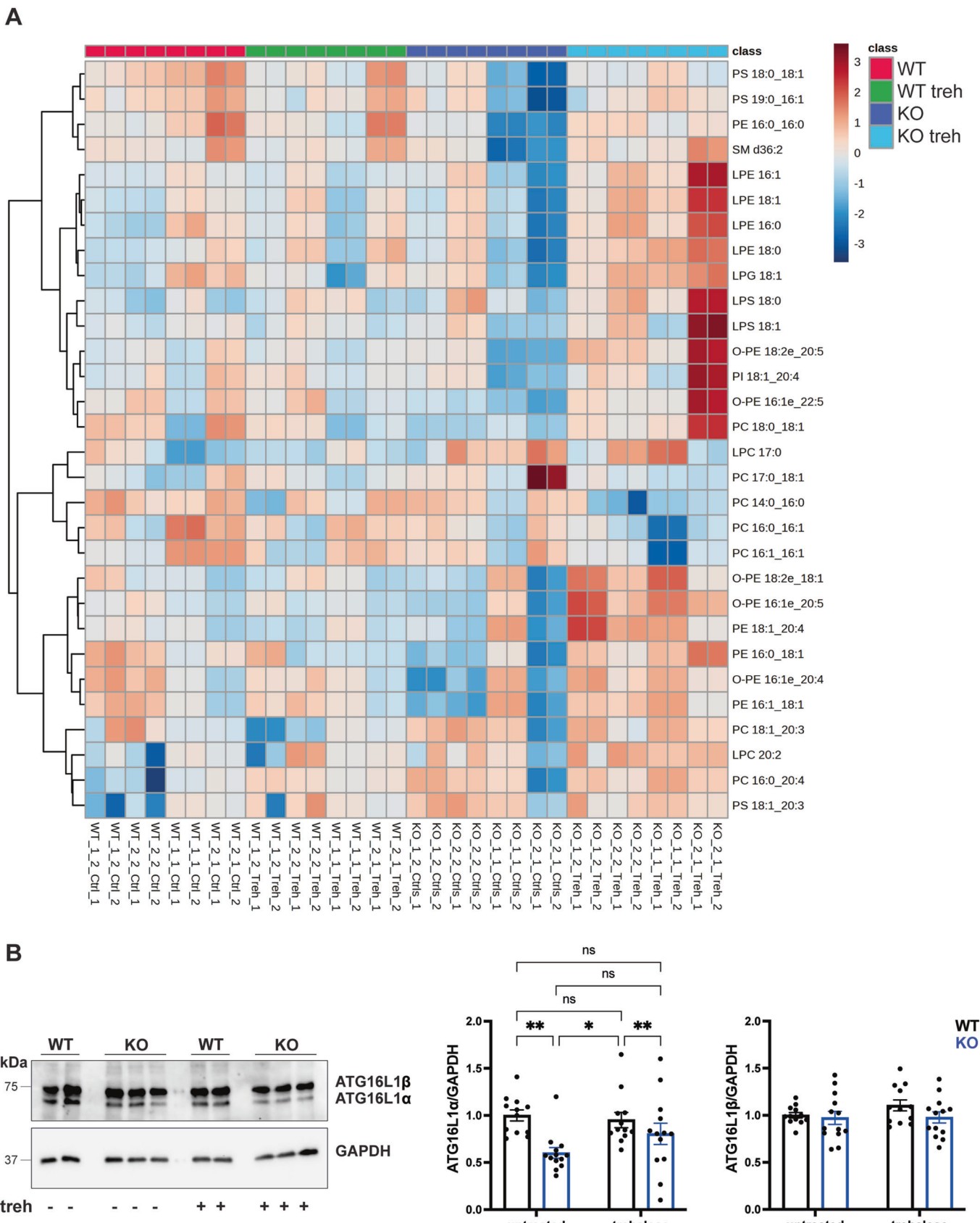

◀ **Figure EV2. ATG16L1 and lipid content of WT and KO treated and untreated neurons.**

(A) Heatmap obtained using the 30 most significant lipids in the complete dataset (ANOVA test). Clustering distance measure: Euclidean. Clustering method for lipids was Ward. PE: diacylglycerophosphoethanolamines; O-PE: alkyl, acylglycerophosphoethanolamines; PS: diacylglycerophosphoserines; PI: diacylglycerophosphoinositols; PC: diacylglycerophosphocholines; SM: ceramide phosphocholines (sphingomyelins); LPG: monoacylglycerophosphoglycerols; LPE: diacylglycerophosphoethanolamines; LPS: monoacylglycerophosphoserines; LPC: monoacylglycerophosphocholines. (B) Representative western blot from lysates of WT and KO cortical neurons (14 DIV) under resting condition or incubated with 25 mM trehalose for 48 h. ATG16L1α and ATG16L1β intensies were quantified by densitometric analysis and normalized on GAPDH intensity. Data were mean ± SEM, normalized on WT ($n = 12$–13 embryos from five independent experiments). Two-way ANOVA with Tukey's multiple comparisons test, $*p < 0.05$, $**p < 0.01$ (ATG16L1α: ut WT vs ut KO $p = 0.0092$, ut WT vs treh WT $p = 0.7545$, ut WT vs treh KO $p = 0.3637$, ut KO vs treh WT $p = 0.0191$, ut KO vs treh KO $p = 0.7545$, treh WT vs treh KO $p = 0.0092$).

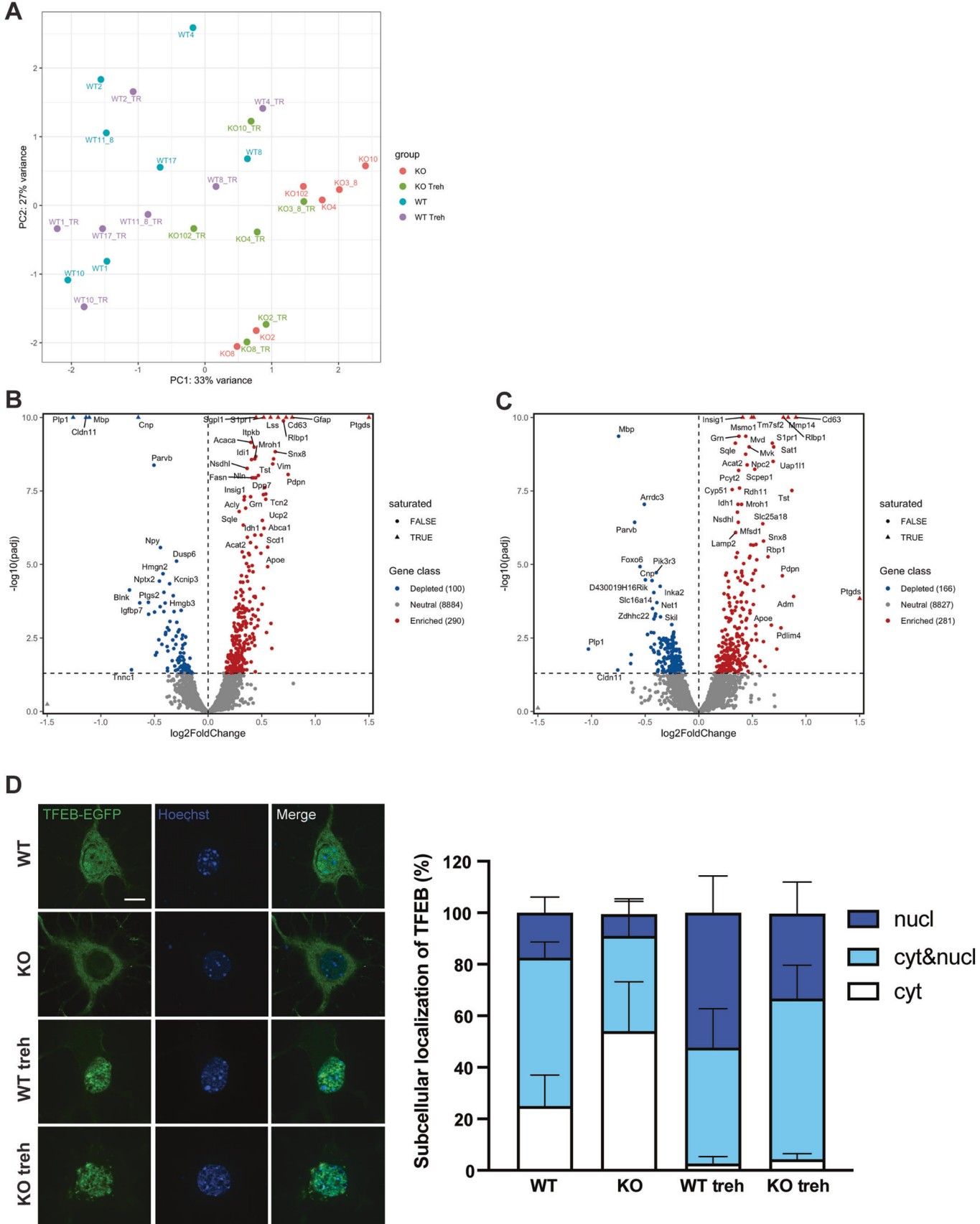

◄ **Figure EV3.  Trehalose administration induces transcriptional changes related to lipidic and auto-lysosomal pathways along with TFEB nuclear translocation.**

(**A**) Principal component analysis (PCA) plot of WT and KO neurons untreated or treated with trehalose. Percentage of variance is reported for both PC1 (first component) and PC2 (second component). (**B, C**) Volcano plot showing the upregulated (red dots) and downregulated (blue dots) DEGs of WT treh vs WT (left) and KO treh vs KO (right) neurons. The x-axis represents the log2 fold change (FC), while the y-axis is the −log 10 (P adj) of RNA-seq data from eight independent biological replicates. The p value was computed using DESeq2 (see Methods). The adjusted p value is obtained using the Benjamini–Hochberg method for correction for multiple hypotheses testing. Differentially expressed genes were assessed using the adjusted p value threshold of 0.05. (**D**) Representative confocal images of WT and KO cortical neurons (14 DIV) transfected with a TFEB-EGFP plasmid, under resting conditions or incubated with 25 mM trehalose for 48 h (treh). GFP immunolabelling and Hoechst nuclear stain are shown. Results are expressed as percentage of cells assigned to each subcellular localization category (either only nuclear, only cytosolic, or nuclear + cytosolic), mean ± SEM (n > 10 cells analyzed/condition for each experiment, from 3 to 4 independent experiments); Scale bar: 10 µm.

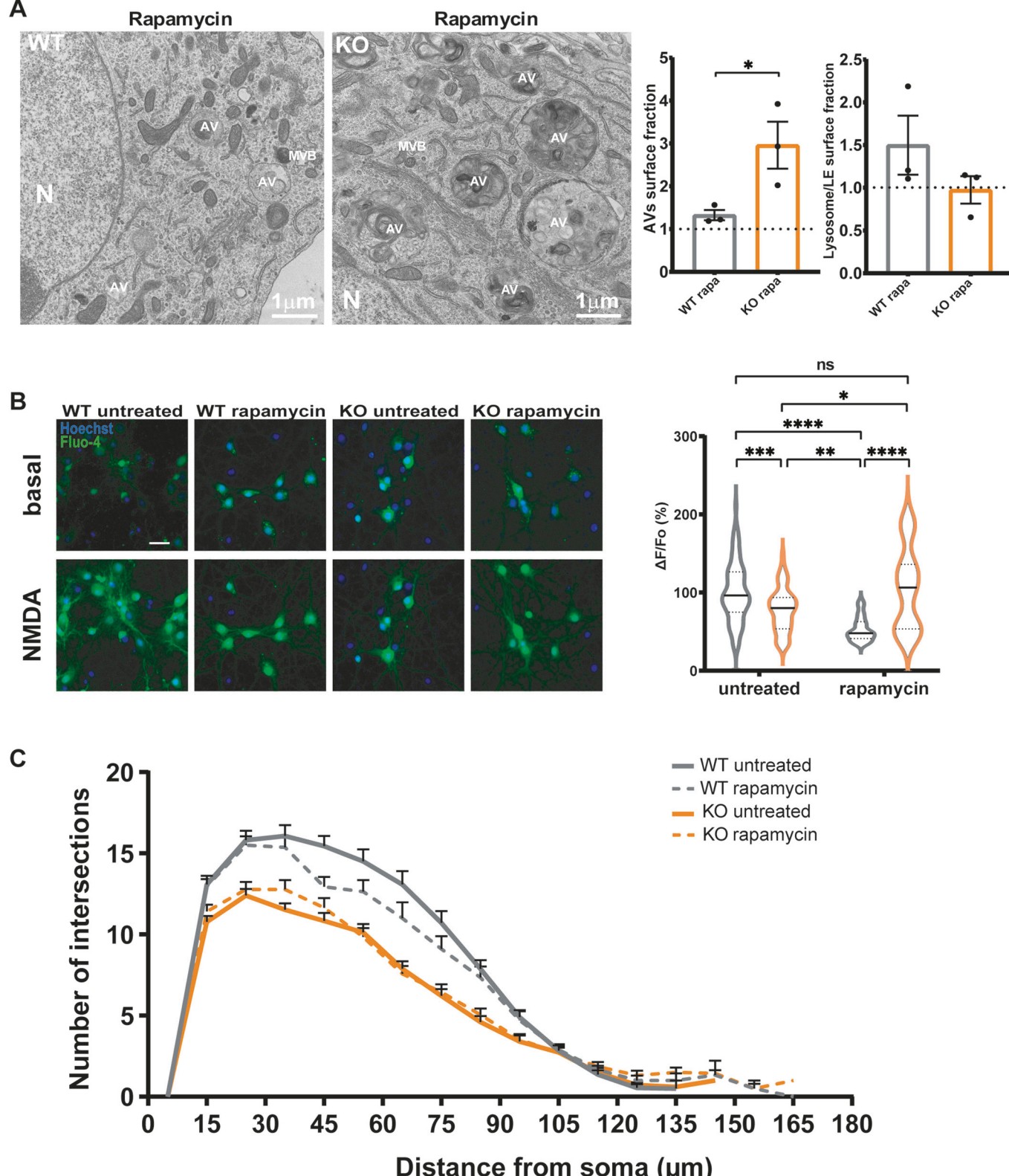

◀ **Figure EV4. Rapamycin enhances autophagic vacuole content and neuronal activity in Mecp2 KO neurons.**

(A) Representative TEM micrographs of WT and KO cortical neurons (14 DIV) incubated with 100 nM rapamycin for 48 h. The area occupied by autophagosomes and autolysosomes (AVs) or by lysosomes over the total cell area in the image was quantified. Data were expressed as mean ± SEM normalized on WT untreated (dashed line on the graph) ($n = 3$ independent experiments, with 58 WT and 61 KO rapa-treated neurons analyzed). N nucleus, AV autophagosome, *=enlarged lysosomes. Unpaired *T*-test, *$p < 0.05$ (Avs surface fraction: rapa WT vs rapa KO $p = 0.0437$; Lysosome/ LE surface fraction: WT rapa vs KO rapa $p = 0.2410$). (B) Representative images of WT and KO primary cortical neurons (14 DIV) loaded with Fluo-4 and exposed to 100 μM NMDA. Scale bar: 40 μm. Data indicate the Fluo-4 intensity (ΔF/F0) of WT and KO treated neurons (violin plots, median ± quartiles; $n = 105$ cells for WT untreated, $n = 32$ cells for WT rapamycin, $n = 63$ cells for KO untreated and $n = 20$ cells for KO rapamycin). Two-way ANOVA followed by Tukey's multiple comparisons test *$p < 0.05$, **$p < 0.01$, ***$p < 0.001$, ****$p < 0.0001$ (ut WT vs ut KO $p = 0.0004$, ut WT vs rapa WT $p < 0.0001$, ut WT vs rapa KO $p = 0.9243$, ut KO vs rapa WT $p = 0.0052$, ut KO vs rapa KO $p = 0.0116$, rapa WT vs rapa KO $p < 0.0001$). (C) The graph reports the average number of intersections measured by Sholl analysis every 15 μm from the soma of WT untreated ($N = 30$ from 2 embryos) or treated ($N = 48$ from 2 embryos) and KO untreated ($N = 30$ from 2 embryos) or treated ($N = 48$ from 2 embryos) neurons. Error bars indicate ±SEM.

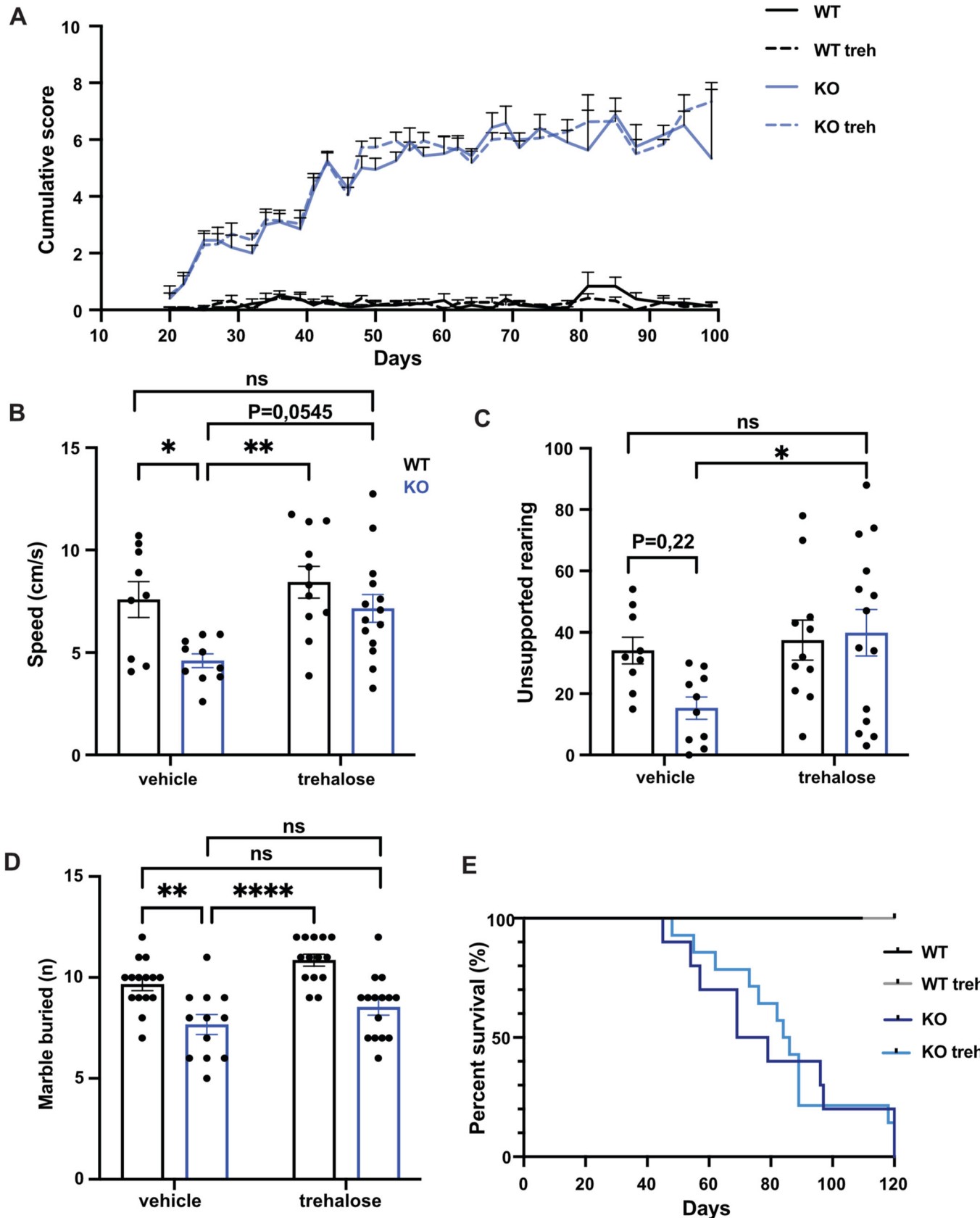

◀ **Figure EV5. Behavioral tests and lifespan analysis in WT and *Mecp2* KO mice upon trehalose treatment.**

(A) Cumulative score. (B) Speed. (C) Unsupported rearing. (D) Marble burying assay. (E) Kaplan–Mayer survival curve of the four experimental groups: WT untreated ($N = 9$) or treated ($N = 11$), and Mecp2-KO untreated ($N = 10$) or treated ($N = 14$). For Marble analysis, $n = 15$ WT vehicle, $12 = $ KO vehicle, $14 = $ WT trehalose, $15 = $ KO trehalose. Bars represent mean values with ± SEM. Behavioral data for **B–D** were analyzed using two-way ANOVA followed by Tukey's multiple comparisons test *$p < 0.05$, **$p < 0.01$, ***$p < 0.001$, ****$p < 0.0001$ (Speed: ut WT vs ut KO $p = 0.0386$, ut WT vs treh WT $p = 0.8493$, ut WT vs treh KO $p = 0.9714$, ut KO vs treh WT $p = 0.0029$, ut KO vs treh KO $p = 0.0545$, treh WT vs treh KO $p = 0.5258$; Unsupported rearing: ut WT vs ut KO $p = 0.2223$, ut WT vs treh WT $p = 0.9844$, ut WT vs treh KO $p = 0.9177$, ut KO vs treh WT $p = 0.0894$, ut KO vs treh KO $p = 0.0349$, treh WT vs treh KO $p = 0.9918$; Marble: ut WT vs ut KO $p = 0.0032$, ut WT vs treh WT $p = 0.1181$, ut WT vs treh KO $p = 0.1366$, ut KO vs treh WT $p > 0.0001$, ut KO vs treh KO $p = 0.3951$, treh WT vs treh KO $p = 0.0003$). Statistics for the survival curves were performed by log-rank test.

