## [Peer Review File · EMBO Molecular Medicine]

Unraveling autophagic imbalances and therapeutic insights in Mecp2-deficient models

Alessandro Esposito, Tommaso Seri, Martina Breccia, Marzia Indrigo, Giuseppina De Rocco, Francesca Nuzzolillo, Vanna Denti, Francesca Pappacena, Gaia Tartaglione, Simone Serrao, Giuseppe Paglia, Luca Murru, Stefano de Pretis, Jean-Michel Cioni, Nicoletta Landsberger, Fabrizia Guarnieri, and Michela Palmieri

Corresponding author(s): Michela Palmieri (palmieri.michela@hsr.it), Alessandro Esposito (esposito.alessandro@hsr.it), Fabrizia Guarnieri (fabrizia.guarnieri@in.cnr.it)

Review Timeline:

Submission Date:	22nd Dec 23
Editorial Decision:	7th Feb 24
Revision Received:	31st Jul 24
Editorial Decision:	16th Aug 24
Revision Received:	16th Sep 24
Accepted:	27th Sep 24

Editor: Poonam Bheda

Transaction Report:

7th Feb 2024

Dear Dr. Palmieri,

Thank you for the submission of your manuscript to EMBO Molecular Medicine. We have now received feedback from the three reviewers who agreed to evaluate your manuscript. As you will see from the reports below, the referees acknowledge the interest of the study; however it is unclear whether some of the major conclusions are sufficiently supported by the data and the reviewers have commented on multiple aspects of the manuscript that should be strengthened in a revision. In particular the effects of trehalose on neuronal morphology and behavioral deficits is modest and requires statistical tests to ensure significance. Furthermore the link to autophagy should be strengthened e.g. via testing autophagy inducers other than trehalose and/or electrophysiology measurements. In addition, any claims not fully supported by the data will need to be justified.

Addressing the reviewers' concerns in full in a point-by-point response will be necessary for further considering the manuscript in our journal, and acceptance of the manuscript will entail a second round of review. EMBO Molecular Medicine encourages a single round of revision only and therefore, acceptance or rejection of the manuscript will depend on the completeness of your responses included in the next, final version of the manuscript. For this reason, and to save you from any frustrations in the end, I would strongly advise against returning an incomplete revision. If you would like to discuss further the points raised by the referees, I am available to do so via email or video. Let me know if you are interested in this option.

We are expecting your revised manuscript within three months, if you anticipate any delay, please contact us. When submitting your revised manuscript, please carefully review the instructions that follow below. We perform an initial quality control of all revised manuscripts before re-review; failure to include requested items will delay the evaluation of your revision.

We require:

4) A .docx formatted letter INCLUDING the reviewers' reports and your detailed point-by-point responses to their comments. As part of the EMBO Press transparent editorial process, the point-by-point response is part of the Review Process File (RPF), which will be published alongside your paper.

5) A complete author checklist, which you can download from our author guidelines (<https://www.embopress.org/page/journal/17574684/authorguide#submissionofrevisions>). Please insert information in the checklist that is also reflected in the manuscript. The completed author checklist will also be part of the RPF.

6) Please note that all corresponding authors are required to supply an ORCID ID for their name upon submission of a revised manuscript.

7) It is mandatory to include a 'Data Availability' section after the Materials and Methods. Before submitting your revision, primary datasets produced in this study need to be deposited in an appropriate public database, and the accession numbers and database listed under 'Data Availability'. Please remember to provide a reviewer password if the datasets are not yet public (see <https://www.embopress.org/page/journal/17574684/authorguide#dataavailability>).

This study includes no data deposited in external repositories.

8) For data quantification: please specify the name of the statistical test used to generate error bars and P values, the number (n) of independent experiments (specify technical or biological replicates) underlying each data point and the test used to calculate p-values in each figure legend. The figure legends should contain a basic description of n, P and the test applied. Graphs must include a description of the bars and the error bars (s.d., s.e.m.). Please provide exact p values.

9) Our journal encourages inclusion of *data citations in the reference list* to directly cite datasets that were re-used and obtained from public databases. Data citations in the article text are distinct from normal bibliographical citations and should

directly link to the database records from which the data can be accessed. In the main text, data citations are formatted as follows: "Data ref: Smith et al, 2001" or "Data ref: NCBI Sequence Read Archive PRJNA342805, 2017". In the Reference list, data citations must be labeled with "[DATASET]". A data reference must provide the database name, accession number/identifiers and a resolvable link to the landing page from which the data can be accessed at the end of the reference. Further instructions are available at .

13) Author contributions: CRedit has replaced the traditional author contributions section because it offers a systematic machine readable author contributions format that allows for more effective research assessment. Please remove the Authors Contributions from the manuscript and use the free text boxes beneath each contributing author's name in our system to add specific details on the author's contribution. More information is available in our guide to authors.

Please also suggest a striking image or visual abstract to illustrate your article as a PNG file 550 px wide x 300-600 px high. Share synopsis text and image, as well as eTOC:

Please note that these would be the final versions and changes during proofing are usually not allowed

16) As part of the EMBO Publications transparent editorial process initiative (see our policy here: https://www.embopress.org/transparent-process#Review_Process), EMBO Molecular Medicine will publish online a Peer Review File (PRF) to accompany accepted manuscripts.

In the event of acceptance, this file will be published in conjunction with your paper and will include the anonymous referee reports, your point-by-point response and all pertinent correspondence relating to the manuscript. Let us know whether you agree with the publication of the PRF and as here, if you want to remove or not any figures from it prior to publication.

I look forward to receiving your revised manuscript.

Yours sincerely,

Poonam Bheda

Poonam Bheda, PhD
Scientific Editor
EMBO Molecular Medicine

**** Reviewer's comments ****

Referee #1 (Comments on Novelty/Model System for Author):

Some key results (Fig1-3) derive from Western blots that are difficult to quantify (due to overexposure and likely saturation of the signal), which makes the corresponding statistical analysis difficult to assess. As to the medical impact, the treatment suggested (trehalose) makes little difference to the general Rett phenotype in the mice.

Referee #1 (Remarks for Author):

In this manuscript, Esposito and colleagues explore alterations in the autophagic process using mouse models of Rett syndrome, both in vitro with primary neurons and in mice. The study addresses an aspect insufficiently explored in the field, despite some previous works suggesting that autophagy is impaired and may represent a relevant pathway underlying important disease features. The work sheds light on the lipidic imbalance in Mecp2-KO cells and its relation to insufficient LC3B-II levels and impaired autophagosome maturation. Notably, trehalose treatment is shown to restore neuronal morphology and rescue some motor deficits and anxiety features in KO mice.

While the manuscript provides novel insights into the dysregulation of autophagy in Rett syndrome and sets the basis for further investigation of this angle, the mechanistic aspect is only preliminarily explored, lacking detailed insights. Key connections, such as the link between dysregulation of ATG16L1 and LC3B, or upstream causes of autophagy dysregulation in connection to Mecp2 mutation, are not adequately addressed. From a translational standpoint, the work lacks evidence of a more substantial and clinically relevant impact of the proposed treatment on Rett Syndrome. The observed effects in the mouse model appear to be modest, while a more pronounced therapeutic outcome is typically expected for manuscripts that propose new therapeutic avenues. For example, the manuscript does not demonstrate improvements in the typical "Bird" score analysis or overall lifespan of the animals post-trehalose treatment. Altogether, I suggest a much deeper exploration of either the mechanistic or the treatment angle be researched before I can recommend publication in EMBO Molecular Medicine.

Major points:

1. It is stated that amino acid starvation enhances autophagic flux, which is generally true. However, the assertion that this is evidenced by a reduction in LC3B-II levels is confusing. Typically, LC3B-II levels increase upon the formation of autophagosomes, indicating induction of autophagy. In addition, the Western blot images in Fig 2B are challenging to interpret due to high signal and variation of the GAPDH control. Considering this issue in multiple Western blots throughout the manuscript, I recommend employing additional methods to assess autophagic flux (e.g., tandem fluorescent-tagged LC3, as demonstrated in Leeman et al., Science. 2018 PMID: 29590078).
2. Does trehalose impact the levels of ATG16L1? Addressing this point would provide a more comprehensive understanding of the treatment effects.
3. Fig 4D and Suppl. Fig 4: A comparison between the levels of PE species achieved in KO+trehalose compared to WT cells is necessary. This analysis would help assess the extent to which the levels are restored and justify the recovery of the autophagy process.
4. The activity of trehalose-treated KO cells is assessed through Ca²⁺ transients in Fig 6B. However, it remains unclear whether these transients are significantly restored to WT levels based on the quantitation shown in the figure. Additionally, Fig. 6C indicates alterations in the density of synaptic vesicles and their docking in KO-treated cells, but a direct link to synaptic activity is not established. To confirm this aspect of the treatment, I recommend providing additional electrophysiological measurements.
5. The manuscript suggests that trehalose treatment leads to phenotypic recovery in KO mice, particularly in motor, exploratory, and anxiety behavior. However, concerning the exploratory behavior (Fig 7E), it appears that trehalose does not significantly improve the marble test, despite the claim that "mutant mice manifested less ability to bury beads, which was improved by trehalose treatment" (line 268). Please clarify this point, as the graph does not show a significant difference between untreated and treated KO animals.

Minor points:

1. I could not find mention of Suppl. Fig. 7D (lifespan of mice upon trehalose treatment) in the text.

Referee #2 (Remarks for Author):

In this paper the authors show that the autophagy process is reduced in Mecp2 KO neurons probably due to phosphatidylethanolamine deficiency. They suggest that trehalose restores lipid imbalance, LC3B lipidation and, consequently, autophagy flux in Mecp2 KO cells. Moreover, the authors claim that trehalose administration, as an autophagy inducer, improves exploratory and locomotor skills of Mecp2 KO mice. Given that loss-of-function mutations are associated to Rett syndrome, the authors suggest that enhancing autophagy could be a potential therapy for treating such neuronal disease. Overall, the effects reported here are in some cases not convincing and I have some concerns about some aspects of this manuscript:

- The decrease in autophagy observed in Mecp2 KO neurons is mild and is only observed at DIV14. Could this decrease in autophagy really be the reason of the phenotypic disorders observed in these neurons (and in Rett syndrome)? Is autophagy really reduced in Mecp2 KO deficient or mutated cells? The levels of P62 and LC3B in Figure 1C give contradictory results, therefore it is impossible to conclude whether autophagy is reduced or not in these Mecp2 mutated primary cells. Further experiments to measure autophagy flux are required in the different models.

Moreover, assessing the LC3II/LC3I ratio is not the most reliable method for measuring autophagy (see Guidelines for the use and interpretation of assays for monitoring autophagy).

Fig1D: The reduced numbers of autophagic vesicles in KO neurons is quite obvious on the displayed EM pictures. The authors could perform additional EM analysis at earlier (or later) DIV. Perhaps AV will be reduced at these time points as well in KO neurons. On these pictures it seems that there are less mitochondria in KO cortical neurons. Can the authors confirm this?

- Fig6B: On the representative images, it is impossible to see any difference in fluorescence between wild-type and KO neurons and the graph shows as well any difference between +/- trehalose in KO neurons. So how can the authors conclude that "the analysis of intracellular Ca²⁺ transients induced by 100 μM NMDA indicated an impaired responsiveness in neurons lacking Mecp2 at 14 DIV. Trehalose administration normalized their ability to respond to NMDA".

- Fig6C: Indicate please the different zones on the EM pictures. This will be easier for nonspecialists to understand.

- Fig7C, D and E: Statistical comparisons should be performed between vehicle and trehalose. A statistical difference between vehicle and trehalose in the same genotype (i.e KO) should be obtained in order to be able to conclude that trehalose restores phenotypic disorders observed in Mecp2 KO mice.

- Another major point is that the authors should repeat some key experiments with an autophagy inducer other than trehalose.

Referee #3 (Comments on Novelty/Model System for Author):

See my comments.

Referee #3 (Remarks for Author):

In this manuscript by Esposito et al, the authors investigated how is the autophagic pathway affected in Rett syndrome (RTT), a neurodevelopmental disorder caused by loss-of-function mutations of MECP2 and often observed in girls. The project started with an intriguing observation that "electron microscopy data back in the '90s indicated the accumulation of undegraded materials in the brain of RTT patients". They first examined the components of autophagic pathway using neurons derived from wild type or RTT mouse embryos, and found lipidation of LC3B-II was reduced in RTT neurons to impact the biogenesis of autophagosomes. A series of following experiments suggests this defect is not due to lysosomal degradation or nutrient sensing by the mTORC1 pathway. Mechanistically they reported the downregulation of ATG16L1 as well as the phosphatidylethanolamine (PE) level in RTT neurons. Furthermore, they showed trehalose, a natural disaccharide, can rescue the reduction of lipidation of LC3B-II, normalize the lysosome/LE surface ratio and PE level. Interestingly, trehalose can also rescue the neuronal defects of RTT neurons including branching, response to glutamatergic stimulus, and the density of synaptic vesicles. Last, they administered trehalose to RTT mice and demonstrated a rescue on motor function, but not other well-documented defects such as shortened life span. Overall, this is a well-designed study to report the less characterized metabolic defect in RTT with some motor defect rescue by trehalose treatment of RTT mice. Although the rescue at animal model is rather limited and moderate, it may not be surprising given the metabolic defect only represents one aspect of RTT. I have a few suggestions for the authors to consider as listed below.

Major points:

1. As demonstrated in Fig S1A, the defective lipidation of LC3B-II only occurred at certain stages during development. Therefore, it is critical to include molecular marks to show the differentiation stage of these DIV neurons as a control to ensure the comparison of WT and RTT neurons at the same developmental stage in Fig 1.

2. For Fig 5, it is nice to see the RNA-seq analysis of WT and RTT with and without trehalose treatment. Could the authors include a PCA analysis of these neurons to evaluate the rescue at the transcriptome level?

3. For RTT mice work, it is not clear whether male or female RTT mice were used. If only male mice were used, a brief discussion on how this trehalose treatment will impact the female mice will be great as majority of RTT patients are girls.
4. For the behavior assay, could the authors include a statistical analysis of KO neurons with and without trehalose treatment? Are the effects from this comparison significant in Fig 7?
5. Last, can this trehalose treatment be applied to human RTT neurons for rescue?

Minor points:

1. LC3B-II is not a common term for general readers. Could the authors spell it out to increase readability in the abstract?
2. A better description on how the SV density quantification is done in Fig 6.
3. It is understandable the authors tried to emphasize the importance of motor defect rescue by trehalose treatment of RTT mice. However, it is not accurate to state in Discussion that "it is important to mention that RTT patients generally do not suffer from a significantly reduced life-span". The shortened life-span is well-documented in RTT literature.

Letter to the Reviewers:

We thank the reviewers for taking the time to evaluate our manuscript and for their insightful comments. Following their observations, we have provided new experiments and added/modified the text accordingly. We believe that our revised version should clarify any concern and has significantly improved the manuscript. A point-by-point answer to their queries and suggestions is provided below.

Referee #1*Comments on Novelty/Model System for Author:*

Some key results (Fig1-3) derive from Western blots that are difficult to quantify (due to overexposure and likely saturation of the signal), which makes the corresponding statistical analysis difficult to assess as to the medical impact, the treatment suggested (trehalose) makes little difference to the general Rett phenotype in the mice.

We apologize that Western blotting experiments throughout the paper are difficult to interpret. All Western blotting experiments were acquired using the Biorad ChemiDoc system coupled with the ImageLab software, which guarantees that we never quantified saturated data. As the Reviewer suggested, we substituted GAPDH panels throughout the paper as well as the entire Fig 2C.

Concerning the second comment, it must be considered that MeCP2 is a very complex multifunctional protein involved in several cellular pathways including protein synthesis, neuronal maturation, synaptic function, lipid metabolism, etc. (PMID: 36835623, PMID: 31629770, PMID: 38391695). As consequence of its loss, all these processes are dysregulated and contribute to the pathogenesis of RTT. In our study we found that LC3B-II lipidation and autophagosome maturation are affected in *Mecp2* KO neurons, therefore our pharmacological strategy aimed at restoring these specific molecular defects through trehalose administration. Indeed, we observed a complete rescue of dendritic complexity and synaptic density in *Mecp2* KO-treated neurons as well as a partial recovery of neuronal activity. At behavioral level, we observed an improvement of locomotor and exploratory phenotypes in KO mice that received trehalose, while the overall bird scoring was not affected. Therefore, our idea is that trehalose in combination with other drugs, with different mechanisms of action, might be used to tackle the disease from more than one side. We added a comment on this in the Discussion (lines 358-359).

Remarks for Author:

In this manuscript, Esposito and colleagues explore alterations in the autophagic process using mouse models of Rett syndrome, both in vitro with primary neurons and in mice. The study addresses an aspect insufficiently explored in the field, despite some previous works suggesting that autophagy is impaired and may represent a relevant pathway underlying important disease features. The work sheds light on the lipidic imbalance in Mecp2-KO cells and its relation to insufficient LC3B-II levels and impaired autophagosome maturation. Notably, trehalose treatment is shown to restore neuronal morphology and rescue some motor deficits and anxiety features in KO mice.

While the manuscript provides novel insights into the dysregulation of autophagy in Rett syndrome and sets the basis for further investigation of this angle, the mechanistic aspect is only preliminarily explored, lacking detailed insights. Key connections, such as the link between dysregulation of ATG16L1 and LC3B, or upstream causes of autophagy dysregulation in connection to Mecp2 mutation, are not adequately addressed. From a translational standpoint, the work lacks evidence of a more substantial and clinically relevant impact of the proposed treatment on Rett Syndrome. The observed effects in the mouse model appear to be modest, while a more pronounced therapeutic outcome is typically expected for manuscripts that propose new therapeutical avenues. For example, the manuscript does not demonstrate improvements in the typical "Bird" score analysis or overall lifespan of the animals post-trehalose treatment. Altogether, I suggest a much deeper exploration of either the mechanistic or the treatment angle be researched before I can recommend publication in EMBO Molecular Medicine.

We thank the Reviewer for recognizing the novel insights of our work as it sheds light on autophagy dysregulation in RTT. We agree that further investigation on the connection between MeCP2 mutations and autophagy dysregulation will be important to increase our comprehension on MeCP2 function.

From the mechanist side, our results indicate that the reduced lipidation of LC3B-II observed in RTT neurons is not attributable to defective lysosomal degradation or nutrient sensing by the mTORC1 pathway. To substantiate the functionality of lysosomes and autophagy flux, we included the tandem fluorescent-tagged LC3 assay, according to Reviewer's suggestion. This additional assay confirmed a defect in autophagosome biogenesis, as the total number of autophagosomes (yellow dots) is reduced in Mecp2 KO neurons and in primary cultured fibroblasts derived from a RTT patient (carrying the *MECP2* c.705delG mutation). The similar percentage of yellow/total red dots between WT and KO cells indicate, instead, the absence of overt lysosomal defect. The detailed results are presented in Fig 1E and 1F.

Moreover, we have demonstrated that the reduction of phosphatidylethanolamine (PE) levels in RTT neurons represents a limiting step in the maturation process of autophagosomes. Consequently, by administering ethanolamine or trehalose, which increase PE levels, to neurons

lacking MeCP2, we were able to restore the levels of LC3B-II, as illustrated in Appendix Fig S3 and Fig 4A.

At transcriptional levels, we also found that trehalose enhances molecular pathways related to membrane lipid and sphingolipid metabolic processes in KO neurons (Fig 5B). In particular, the sphingosine-1-phosphate-lyase 1, Sgpl1, the enzyme that catalyzes the conversion of sphingosine 1-phosphate (S1P) in phosphoethanolamine, was increased following trehalose treatment (Appendix Table S2). Interestingly, the S1P pathway is a crucial mechanism for neuronal autophagy because provides PE for LC3 conjugation (PMID: 28521611, PMID: 26477494). Accordingly, another actor in this process, the sphingosine 1 phosphate receptor 1 (S1PR1) was also upregulated by trehalose (Appendix Table S2).

In our study we finally observed a reduction of ATG16L1 α in neurons lacking *Mecp2*, while no differences were detected for the β isoform. These two isoforms are mostly expressed in mammals and have been described to localize at phagophore where mediate the recruitment of Atg12-Atg5 complex, thereby specifying the site for LC3 lipidation (PMID: 18321988). However, the exact mechanism for this recruitment is not fully understood, nor the specific role of these isoforms, although a recent report suggested that isoform β is probably involved in the lipidation of LC3 on damaged endosomes (PMID: 30778222).

Thanks to Reviewer's suggestion we thus evaluated whether trehalose could restore the levels of ATG16L1 α isoform in KO neurons, but this is not the case, further suggesting that PE deficiency might be the primary cause of defective LC3B lipidation. These results are now included in Fig EV2B.

Concerning the moderate therapeutic outcomes of trehalose obtained *in vivo*, we would like to mention that trehalose was administered intraperitoneally every other day for only 21 days starting from P18, and this was sufficient to ameliorate the locomotor and exploratory defects of KO mouse model. In addition, the survival curve trend indicates that KO-treated animals showed a slower progression of the disease with 50% of them living 15 days longer than their untreated littermates (Fig EV5E). Overall trehalose ameliorated their lives of about 30 days after the last injection.

Clearly, a more detailed pharmacological study will be needed in the future to define what is the most efficacious therapeutic regimen to be applied for trehalose (e.g., more frequent administrations, more prolonged treatment), but we are convinced we provided a first proof-of-principle that this molecule (or other similar drugs) might be worth studying in the context of RTT.

Major points:

1. It is stated that amino acid starvation enhances autophagic flux, which is generally true. However, the assertion that this is evidenced by a reduction in LC3B-II levels is confusing. Typically, LC3B-II levels increase upon the formation of autophagosomes, indicating induction of

autophagy. In addition, the Western blot images in Fig 2B are challenging to interpret due to high signal and variation of the GAPDH control. Considering this issue in multiple Western blots throughout the manuscript, I recommend employing additional methods to assess autophagic flux (e.g., tandem fluorescent-tagged LC3, as demonstrated in Leeman et al., *Science*. 2018 PMID: 29590078).

We thank the Reviewer for this observation, and we understand that an enhancement of autophagy flux evidenced by a reduction in LC3B-II levels upon amino acid starvation as shown in Fig 2B might be counterintuitive. However, in our experiments, we consistently detected a reduction of both p62 and LC3B-II in WT cortical neurons deprived of nutrients, as shown in the figure below. This is in line with other reports in the literature that detect a reduction in LC3B-II upon induction of autophagic flux, as LC3B-II is itself a substrate of autophagic degradation (see for example Napolitano G, et al. *EMBO Mol Med*. 2015;7(2):158-174.; Nowosad A, et al. *Nat Cell Biol*. 2020;22(9):1076-1090). In addition, the “Guidelines for the use and interpretation of assays for monitoring autophagy, 4th edition” at Fig. 7C report that the outcome of starvation on the levels of LC3B-II may differ depending on the cell type (e.g., while LC3B-II levels increase upon starvation in HEK293 cells, they decrease in HeLa), and may also depend on the time point of analysis after starvation induction.

As mentioned above, we agree with the Reviewer that the autophagic deficit in MeCP2 KO neurons needed to be strengthened by additional assays. As suggested by the Reviewer, we performed the mCherry-EGFP-LC3 tandem assay, which clearly prove a defect in autophagosome biogenesis assessed by the lower number of yellow dots in *Mecp2* KO neurons and in RTT patient fibroblasts. The similar percentage in the ratio of yellow/total red dots between WT and KO cells, instead, indicates the absence of overt lysosomal deficit. The detailed results are now included in Fig 1E and 1F.

2. Does trehalose impact the levels of ATG16L1? Addressing this point would provide a more comprehensive understanding of the treatment effects.

We thank the Reviewer for raising this interesting point; as mentioned above, we analyzed the levels of ATG16L1 upon trehalose treatment and we did not observe a recovery of ATG16L1 α levels in *Mecp2* KO neurons, further suggesting that PE deficiency might be the primary cause of defective LC3B lipidation. ATG16L1 α and β isoforms recruit the ATG12-5 complex to phagophore with consequent lipidation of LC3-II, however the molecular mechanism how this process take place is still not fully investigated. The results are now included in Fig EV2B.

3. Fig 4D and Suppl. Fig 4: A comparison between the levels of PE species achieved in KO+trehalose compared to WT cells is necessary. This analysis would help assess the extent to which the levels are restored and justify the recovery of the autophagy process.

We agree with the Reviewer that direct comparison between WT, KO and KO+trehalose in terms of PE species modulation is highly informative. We inserted new graphs including WT, KO and KO+trehalose samples in Fig 4D. However, since data would have been duplicated in this Figure and in Fig 3C, we decided to remove the latter in the revised version of the manuscript.

4. The activity of trehalose-treated KO cells is assessed through Ca²⁺ transients in Fig 6B. However, it remains unclear whether these transients are significantly restored to WT levels based on the quantitation shown in the figure.

We thank the Reviewer for this observation, we clarify that KO cells treated with trehalose are not significantly different from the untreated ones. However, we want to specify that they are also no more significantly different from WT neurons, meaning that trehalose partially rescued the calcium influx in neurons lacking *Mecp2*. We modify the text accordingly (line 253-254).

Additionally, Fig. 6C indicates alterations in the density of synaptic vesicles and their docking in KO-treated cells, but a direct link to synaptic activity is not established. To confirm this aspect of the treatment, I recommend providing additional electrophysiological measurements.

To correlate morphological and functional findings, we performed electrophysiological recordings of miniature excitatory post-synaptic currents (mEPSCs) in WT and KO neurons treated with trehalose. KO neurons displayed a basal increase in event amplitude and frequency, as already reported in hippocampal cultures by Li et al. (Li W, et al., *PNAS*. 2016;113(11):E1575-E1584),

even though contrasting results have been published by others (e.g., Nelson ED, et al. *Curr Biol.* 2006;16(7):710-716; Chao HT, et al. *Neuron.* 2007;56(1):58-65). Trehalose treatment did not rescue these phenotypes, despite the recovery in the density of synaptic vesicles observed by TEM. However, mEPSC amplitude and frequency depend on many factors, including the expression levels of AMPAR, the number of synapses formed, the probability of vesicle release, the function of presynaptic receptors such as mGluRs, which are almost invariably known to be altered in *Mecp2* KO neurons and might not be all normalized by trehalose. Results of this additional experiment have been included in Appendix Fig S3C.

5. The manuscript suggests that trehalose treatment leads to phenotypic recovery in KO mice, particularly in motor, exploratory, and anxiety behavior. However, concerning the exploratory behavior (Fig 7E), it appears that trehalose does not significantly improve the marble test, despite the claim that "mutant mice manifested less ability to bury beads, which was improved by trehalose treatment" (line 268). Please clarify this point, as the graph does not show a significant difference between untreated and treated KO animals.

We apologize for this oversight; we agree with Reviewer and moved the Fig7E in Fig EV5D. The text is modified accordingly (lines 296-298).

Minor points:

1. I could not find mention of Suppl. Fig. 7D (lifespan of mice upon trehalose treatment) in the text.

The Supplementary Fig.7D that now is identified as Fig EV5E is mentioned in the manuscript at lines 305-307: "Although we did not find a striking effect on the survival of *Mecp2* KO treated animals, we could observe that 50% of them lived 15 days longer than untreated ones (Fig EV5E)".

Referee #2

Remarks for Author:

*In this paper the authors show that the autophagy process is reduced in *Mecp2* KO neurons probably due to phosphatidylethanolamine deficiency. They suggest that trehalose restores lipid imbalance, LC3B lipidation and, consequently, autophagy flux in *Mecp2* KO cells. Moreover, the authors claim that trehalose administration, as an autophagy inducer, improves exploratory and locomotor skills of *Mecp2* KO mice. Given that loss-of-function mutations are associated to Rett syndrome, the authors suggest that enhancing autophagy could be a potential therapy for treating*

such neuronal disease. Overall, the effects reported here are in some cases not convincing and I have some concerns about some aspects of this manuscript:

1. The decrease in autophagy observed in Mecp2 KO neurons is mild and is only observed at DIV14. Could this decrease in autophagy really be the reason of the phenotypic disorders observed in these neurons (and in Rett syndrome)? Is autophagy really reduced in Mecp2 KO deficient or mutated cells? The levels of P62 and LC3B in Figure 1C give contradictory results, therefore it is impossible to conclude whether autophagy is reduced or not in these Mecp2 mutated primary cells. Further experiments to measure autophagy flux are required in the different models. Moreover, assessing the LC3II/LC3I ratio is not the most reliable method for measuring autophagy (see Guidelines for the use and interpretation of assays for monitoring autophagy).

We thank the Reviewer for raising these interesting questions. As several studies showed the relevance of autophagy catabolic function in the physiology of brain development, we believe that its dysfunction may contribute to the disease phenotypes rather than being the cause. Since Mecp2 is involved in several cellular pathways, it is conceivable (and extensively demonstrated) that its loss will generate a series of dysregulated pathway that together will give rise to Rett syndrome pathology.

As suggested by the Reviewer, to consolidate the autophagic defect in Mecp2 KO neurons, we performed the mCherry-EGFP-LC3 tandem assay. The results clearly demonstrate that the autophagosome biogenesis is impaired in Mecp2 KO neurons and in RTT patient fibroblasts as indicated by the lower number of yellow puncta (autophagosomes) in mutant cells, while the lysosomal function is preserved as measured by the ratio of yellow/red dots. These data are now included in Fig 1E and 1F.

2. Fig1D: The reduced numbers of autophagic vesicles in KO neurons is quite obvious on the displayed EM pictures. The authors could perform additional EM analysis at earlier (or later) DIV. Perhaps AV will be reduced at these time points as well in KO neurons. On these pictures it seems that there are less mitochondria in KO cortical neurons. Can the authors confirm this?

Following the Reviewer's suggestion, we performed an additional TEM analysis on WT and KO neurons at an earlier time point (DIV9). Below are the representative images and quantifications of autophagic vesicles and lysosomes showing no differences between WT and KO neurons.

We did not specifically analyze the number of mitochondria in WT and KO neurons in this study, but mitochondrial dysfunction in Rett syndrome has been reported elsewhere. The picture was anyhow particularly biased, so we substituted the panel (Fig 1D).

3. *Fig6B: On the representative images, it is impossible to see any difference in fluorescence between wild-type and KO neurons and the graph shows as well as any difference between +/- trehalose in KO neurons. So how can the authors conclude that "the analysis of intracellular Ca²⁺ transients induced by 100 μM NMDA indicated an impaired responsiveness in neurons lacking Mecp2 at 14 DIV. Trehalose administration normalized their ability to respond to NMDA".*

We apologize for the quality of images in Fig 6B. We have now substituted the representative pictures. Moreover, as clarified above, the responsiveness of *Mecp2* KO neurons to glutamatergic stimuli are not significantly different either from untreated KO or from untreated WT neurons, meaning to us that trehalose partially normalized their ability to respond to NMDA. The text was modified accordingly (lines 253-254).

4. Fig6C: Indicate please the different zones on the EM pictures. This will be easier for nonspecialists to understand.

We apologize for the lack of details; we have now indicated the different zones on the TEM picture (Fig 6C).

5. Fig7C, D and E: Statistical comparisons should be performed between vehicle and trehalose. A statistical difference between vehicle and trehalose in the same genotype (i.e KO) should be obtained in order to be able to conclude that trehalose restores phenotypic disorders observed in *Mecp2* KO mice.

We thank the Reviewer for this observation, the statistical comparisons between vehicle and trehalose in the same genotype have been added. Also, we would like to take the opportunity to specify that all figures in our manuscript comprise of all four experimental groups. Although trehalose is a very safe and well documented molecule, we decided for transparency to include the WT-treh group in our paradigms and statistics. Indeed, if we would have omitted the WT-treh group from the total distance (Fig. 7D) or from the speed (now Fig EV5B) the statistics achieved by one-way ANOVA would have indicated a *p-value* equal to 0,029 and 0,026 between KO and KO-treated groups, respectively. Below are the graphs showing the results.

6. Another major point is that the authors should repeat some key experiments with an autophagy inducer other than trehalose.

We thank the Reviewer for the excellent point. To corroborate the relevance of our work we repeated some key experiments with the well-studied autophagy inducer rapamycin. In particular we assessed the recovery of autophagosome content and neuronal defects in *Mecp2* KO neurons.

Similarly to trehalose data, we found that rapamycin treatment increased the autophagic vesicles to a larger extent in KO than WT treated neurons (Fig EV4A); moreover, it also stimulated the neuronal activity in *Mecp2* deficient neurons upon 48 hours of administration (Fig EV4B), while it did not improve neuronal morphology (Fig EV4C). Notably, the responsiveness of WT neurons to NMDA stimuli was also significantly affected by rapamycin treatment. Overall, these data together suggest that, depending on their specific mechanism of action, distinct autophagic inducers may differentially impact RTT neuronal features. All results are included in Fig EV4 and described in the lines 265-271 of the manuscript.

Referee #3

Remarks for Author:

In this manuscript by Esposito et al, the authors investigated how is the autophagic pathway affected in Rett syndrome (RTT), a neurodevelopmental disorder caused by loss-of-function mutations of MECP2 and often observed in girls. The project started with an intriguing observation that "electron microscopy data back in the '90s indicated the accumulation of undegraded materials in the brain of RTT patients". They first examined the components of autophagic pathway using neurons derived from wild type or RTT mouse embryos, and found lipidation of LC3B-II was reduced in RTT neurons to impact the biogenesis of autophagosomes. A series of following experiments suggests this defect is not due to lysosomal degradation or nutrient sensing by the mTORC1 pathway. Mechanistically they reported the downregulation of ATG16L1 α as well as the phosphatidylethanolamine (PE) level in RTT neurons. Furthermore, they showed trehalose, a natural disaccharide, can rescue the reduction of lipidation of LC3B-II, normalize the lysosome/LE surface ratio and PE level. Interestingly, trehalose can also rescue the neuronal defects of RTT neurons including branching, response to glutamatergic stimulus, and the density of synaptic vesicles. Last, they administered trehalose to RTT mice and demonstrated a rescue on motor function, but not other well- documented defects such as shortened life span. Overall, this is a well-designed study to report the less characterized metabolic defect in RTT with some motor defect rescue by trehalose treatment of RTT mice. Although the rescue at animal model is rather limited and moderate, it may not be surprising given the metabolic defect only represents one aspect of RTT. I have a few suggestions for the authors to consider as listed below.

We thank the reviewer for all the positive comments on our work and the following constructive comments to improve the manuscript.

Major points:

1. *As demonstrated in Fig S1A, the defective lipidation of LC3B-II only occurred at certain stages during development. Therefore, it is critical to include molecular marks to show the differentiation stage of these DIV neurons as a control to ensure the comparison of WT and RTT neurons at the same developmental stage in Fig 1.*

We thank the Reviewer for the interesting suggestion to verify that the defect of LC3B-II lipidation observed at 14 DIV is not related to differences in the differentiation stage of KO neurons in culture with respect to WT cells. We included the analysis of NeuN, PSD95, SNAP25 and VAMP2 by Western blotting at different stages of development. The results show a similar expression pattern of these markers in WT and KO neurons along development, suggesting that we are correctly comparing similar differentiation windows. Results have been included in Fig EV1B. This is in line with previous work indicating that RTT neurons do not fail to reach differentiation milestones (i.e., they form functional synapses at 14 DIV), even though they display a reduced level of maturation (Scaramuzza et al., *EMBO Mol Med.* 2021;13(4):e12433. doi:10.15252/emmm.202012433; Frasca et al., *EMBO Mol Med.* 2020;12(6):e10270. doi:10.15252/emmm.201910270). The text is added in lines 113-122 of this manuscript.

2. *For Fig 5, it is nice to see the RNA-seq analysis of WT and RTT with and without trehalose treatment. Could the authors include a PCA analysis of these neurons to evaluate the rescue at the transcriptome level?*

This is an excellent point. We have now added the PCA analysis in Fig EV5A, which shows a clear separation between WT and KO neurons and a tendency of KO treated cells towards the WTs. The results are mentioned in the text at lines 217-219.

3. *For RTT mice work, it is not clear whether male or female RTT mice were used. If only male mice were used, a brief discussion on how this trehalose treatment will impact the female mice will be great as majority of RTT patients are girls.*

We thank the Reviewer for this observation, we further clarified the gender used in this work and modify the Materials and Methods section accordingly (lines 389-390).

We agree that discussing on how trehalose treatment may impact the female mice is an interesting point. We have added a comment in the Discussion section (lines 372-375).

4. *For the behavior assay, could the authors include a statistical analysis of KO neurons with and without trehalose treatment? Are the effects from this comparison significant in Fig 7?*

We thank the Reviewer for noticing the missing values. We have now corrected the figure and added the statistical analysis between KO vehicle and KO trehalose-treated animals. Importantly, we found statistical significance in Fig. 7B, 7E and 7F.

5. Last, can this trehalose treatment be applied to human RTT neurons for rescue?

We understand the significance of this experiment. Despite we are working on developing the necessary skills and equipment to handle these valuable samples, we currently do not have the expertise to culture human RTT neurons or differentiate iPSCs. Therefore, we are unable to address this aspect at this time. However, it is our interest to test whether trehalose could rescue the alterations seen in neurons derived from RTT-iPSCs including the reduced spine density, the altered calcium signaling etc...(PMID: 21074045). This model will be also useful to screen other autophagy modulators known to impact the autophagosome biogenesis, thereby providing a more translational value to our study. We added a comment on this in the Discussion (lines 375-377).

Minor points:

1. LC3B-II is not a common term for general readers. Could the authors spell it out to increase readability in the abstract?

For clarity, we spelled out LC3B-II at line 57 of the Introduction.

2. A better description on how the SV density quantification is done in Fig 6.

We have updated our Materials and Methods section with more details on how the SV density quantification was done (lines 548-552).

3. It is understandable the authors tried to emphasize the importance of motor defect rescue by trehalose treatment of RTT mice. However, it is not accurate to state in Discussion that "it is important to mention that RTT patients generally do not suffer from a significantly reduced life-span". The shortened life-span is well-documented in RTT literature.

We apologize for our statement and understand the Reviewer's concern, we have modified the sentence in the Discussion section accordingly (lines 369-372).

16th Aug 2024

Dear Dr. Palmieri,

Thank you for the submission of your revised manuscript to EMBO Molecular Medicine. Your manuscript has now been re-reviewed by two of the original reviewers. Based on their advice (included below), I am pleased to inform you that we will be able to accept your manuscript pending the following final amendments and appropriate response to reviewers:

- 1) Data Availability: Please move the Data Availability section to the end of the Methods section
- 2) Data availability: Please release the GSE271893 dataset so that it is publicly available. Please be aware that all deposited datasets should be freely accessible prior to publication.
- 3) Author contributions: Please remove it from the manuscript and specify author contributions in our submission system. CRedit has replaced the traditional author contributions section because it offers a systematic machine-readable author contributions format that allows for more effective research assessment. You are encouraged to use the free text boxes beneath each contributing author's name to add specific details on the author's contribution. More information is available in our guide to authors:
<https://www.embopress.org/page/journal/17574684/authorguide#authorshipguidelines>
- 4) References: Please correct the reference citation in the reference list. Where there are more than 10 authors on a paper, note that only 10 should be listed, followed by "et al.". Please also remove the DOIs. Please check "Author Guidelines" for more information.
<https://www.embopress.org/page/journal/17574684/authorguide#referencesformat>
- 5) In the Methods, please take care of the following:
 - Cell lines: Please be sure to include a sentence in the Methods as to whether or not the cell lines were recently authenticated and tested for mycoplasma contamination and update the Author Checklist with this information.
 - Primers: Please update the Author Checklist to indicate that primers were used and are defined in the Methods ('Animals' section).
- 6) Please place individual sections of the manuscript in the following order: Title page - Abstract & Keywords - Introduction - Results - Discussion - Methods - Data Availability - Acknowledgements - Disclosure and Competing Interests Statement - The Paper Explained - References - Figure Legends - Expanded View Figure Legends.
- 7) For the figures and figure legends, please take care of the following:
 - Please make sure to include a callout for Appendix Figure S1 in the main manuscript.
 - In the figure legends, please correct figure titles to "Figure EV1" etc.; the heading "Expanded View Figure Legends" should stay as is.
 - Please define the annotated p values ******* as well as provide the exact p-values for the same in the figure or legend of figure 6a; EV 4a; as appropriate.
 - Please provide exact p values in the figure or legends for figures 1a-b, d-f; 2a-b; 4a, d; 6b-c; 7b, d-f; EV 1a; EV 2b; EV 4b.
 - Please note that in figures 1b; 3a; EV 1d; EV 4b; EV 5b-d; there is a mismatch between the annotated p values in the figure legend and the annotated p values in the figure file that should be corrected. Also, please provide the exact p-values for the same if applicable.
 - Please note that for the figure EV 4c, p-values and statistical tests are indicated in the legends. However, comparison for the same, ******/****** has not been represented in the figure. Please rectify this in the figure or legend as applicable. Also, please provide the exact p-values for the same if applicable.
 - Please indicate the statistical test used for data analysis in the legends of figures 3b; 4c; 5a-b, d-f; 6a; EV 3b-c.
 - Please note that the box plots need to be defined in terms of bounds of box and whiskers, and percentile in the legends of figures 1a; 2b-c; 3a; EV 1a, d.
 - Please note that the error bars are not defined in the legends of figures EV 4c; EV 5a-d.
 - Please note that the scale bar is missing for figure 6a.
- 8) Appendix file: Please remove the yellow highlighting and add page numbers to the Table of Contents.
- 9) Funding: Please note that funding information should be given in the "Acknowledgements" section (not in its own separate section).
- 10) Synopsis:
 - Synopsis image: Please provide a graphic that summarises the main findings of the manuscript on a glance and upload it as a high-resolution jpeg file 550 pixels wide x (250-400) pixels high.
 - Synopsis text: Please provide a short standfirst (maximum of 300 characters, including space), limit the bullet points to max. 5 and upload it as a separate .doc file. Please write the bullet points to summarise the key NEW findings. They should be designed to be complementary to the abstract - i.e. not repeat the same text. We encourage inclusion of key acronyms and quantitative information (maximum of 30 words / bullet point). Please use the passive voice.
 - Please check your synopsis text and image before submission with your revised manuscript. Please be aware that in the proof stage minor corrections only are allowed (e.g., typos).
- 11) Source Data: Source Data for Fig 4C seems to be missing. In addition, the GAPDH control blot uploaded as Source Data for Figure 2B does not match the figure in the main manuscript. Please check this.
- 12) As part of the EMBO Publications transparent editorial process initiative (see our policy here:

https://www.embopress.org/transparent-process#Review_Process), EMBO Molecular Medicine will publish online a Peer Review File (PRF) to accompany accepted manuscripts. This file will be published in conjunction with your paper and will include the anonymous referee reports, your point-by-point response and all pertinent correspondence relating to the manuscript. Let us know whether you agree with the publication of the PRF and as here, if you want to remove or not any figures from it prior to publication. Please note that the Authors checklist will be published at the end of the PRF.

13) Please provide a point-by-point letter INCLUDING my comments as well as the reviewer's reports and your detailed responses (as Word file).

I look forward to reading a new revised version of your manuscript as soon as possible.

Yours sincerely,

Poonam Bheda

Poonam Bheda, PhD
Scientific Editor
EMBO Molecular Medicine

***** Reviewer's comments *****

Referee #1 (Comments on Novelty/Model System for Author):

To achieve a greater medical impact, a more complete in vivo testing of trehalose (including a longer timeframe of administration) would be desirable, along with the use of a female mouse model of RTT.

Referee #1 (Remarks for Author):

In response to my concerns, the authors have conducted a series of new analyses that have clarified many issues, specifically:

1. To help interpret the changes in LC3I/II in terms of autophagic flux, they now provide the analysis of an LC3 reporter that confirms the defect in autophagosome biogenesis (new Figure 1E, 1F).
2. They have clarified that ATG16L1 levels are not altered by trehalose treatment.
3. They now provide a measure of the levels of PE species in the treated samples compared to WT cells.
4. They have provided the required quantifications in electrophysiological assays upon trehalose treatment. Additionally, they have performed recordings of mEPSCs to complement the synaptic activity assessment of trehalose treatment. Although these experiments have not shown a significant change in the phenotype, I appreciate the effort in carrying out these assays and consider them useful for providing a clearer picture of the extent of phenotypic reversal with trehalose.

Other minor points have also been properly addressed. I consider the manuscript much improved and better suited for publication.

Referee #3 (Comments on Novelty/Model System for Author):

Only male mouse model for Rett syndrome was used in this study. Human RTT neurons derived from iPSC/ESC were recommended but not used in the revised study.

Referee #3 (Remarks for Author):

The authors performed a few new experiments to address some raised questions from the review process. It is understandable that human RTT iPSC and derived neurons might not be available in their lab. Lack of result on the effect by the trehalose treatment on female RTT mice dampened my enthusiasm for this study, as majority of RTT patients are girls.

2nd Author's Response to Reviewers

5th Sep 2024

Referee #1 (Comments on Novelty/Model System for Author):

To achieve a greater medical impact, a more complete in vivo testing of trehalose (including a longer timeframe of administration) would be desirable, along with the use of a female mouse model of RTT.

We understand the Referee comment and we agree that longer time-frame of administration and use of a female mouse model of RTT will have contributed to a greater medical impact of the study. However, as also explained to Referee #3, our Minister of Health did not give us the permission of using both genders, thus we had to utilize only the male animals in this work.

Referee #1 (Remarks for Author):

In response to my concerns, the authors have conducted a series of new analyses that have clarified many issues, specifically:

- 1. To help interpret the changes in LC3/II in terms of autophagic flux, they now provide the analysis of an LC3 reporter that confirms the defect in autophagosome biogenesis (new Figure 1E, 1F).*
- 2. They have clarified that ATG16L1 levels are not altered by trehalose treatment.*
- 3. They now provide a measure of the levels of PE species in the treated samples compared to WT cells.*
- 4. They have provided the required quantifications in electrophysiological assays upon trehalose treatment. Additionally, they have performed recordings of mEPSCs to complement the synaptic activity assessment of trehalose treatment. Although these experiments have not shown a significant change in the phenotype, I appreciate the effort in carrying out these assays and consider them useful for providing a clearer picture of the extent of phenotypic reversal with trehalose.*

Other minor points have also been properly addressed. I consider the manuscript much improved and better suited for publication.

We thank the Referee #1 and we are glad that we have been able to improve the manuscript.

Referee #3 (Comments on Novelty/Model System for Author):

Only male mouse model for Rett syndrome was used in this study. Human RTT neurons derived from iPSC/ESC were recommended but not used in the revised study.

Referee #3 (Remarks for Author):

The authors performed a few new experiments to address some raised questions from the review process. It is understandable that human RTT iPSC and derived neurons might not be available in their lab. Lack of result on the effect by the trehalose treatment on female RTT mice dampened my enthusiasm for this study, as majority of RTT patients are girls.

We thank Referee #3 for the input to the manuscript. We understand that the use of human RTT neurons derived from iPSC/ESC or the addition of *in vivo* studies in female RTT mice would have improved the translational impact of our study, however, as the Referee might know, unfortunately the Italian Government has very restricted rules regarding the use of animal models for experimentation. Indeed, when we asked for permission on both genders for this study, the Italian Minister of Health only agreed for the use of male mice. Therefore, our interest is now to first confirm the validity of trehalose in human RTT neurons derived from iPSC. We will employ cell lines expressing the mutant allele alone or expressing the WT and mutant alleles together, thus mimicking the patient conditions. Confident for positive results, and certain of our male mice's data, we will ask the Italian Minister of Health for the possibility to perform the trehalose study on HET females. We hope that the Referee can understand the difficulty of accomplishing this set of experiments in a proper time window and consider our work for publication as we answered all requests that were in our possibilities to address.

We agree with the publication of the Peer Review File.

27th Sep 2024

Dear Dr. Palmieri,

Congratulations on an excellent manuscript, I am pleased to inform you that your manuscript has been accepted for publication in the EMBO Molecular Medicine. Thank you for your comprehensive response to referee concerns and for providing detailed source data. It has been a pleasure to work with you to get this to the acceptance stage.

Yours sincerely,

Poonam Bheda, PhD
Scientific Editor
EMBO Molecular Medicine
